# GRAMMAR REINFORCEMENT LEARNING: PATH AND CYCLE COUNTING IN GRAPHS WITH A CONTEXT-FREE GRAMMAR AND TRANSFORMER APPROACH

**Jason Piquenot**[1]        **Maxime Bérar**[1]        **Pierre Héroux**[1]        **Jean-Yves Ramel**[2]

**Romain Raveaux**[2]        **Sébastien Adam**[1]

[1]LITIS Lab, University of Rouen Normandy, France
[2]LIFAT Lab, University of Tours, France

## ABSTRACT

This paper presents Grammar Reinforcement Learning (GRL), a reinforcement learning algorithm that uses Monte Carlo Tree Search (MCTS) and a transformer architecture that models a Pushdown Automaton (PDA) within a context-free grammar (CFG) framework. Taking as use case the problem of efficiently counting paths and cycles in graphs, a key challenge in network analysis, computer science, biology, and social sciences, GRL discovers new matrix-based formulas for path/cycle counting that improve computational efficiency by factors of two to six w.r.t state-of-the-art approaches. Our contributions include: (i) a framework for generating gramformers that operate within a CFG, (ii) the development of GRL for optimizing formulas within grammatical structures, and (iii) the discovery of novel formulas for graph substructure counting, leading to significant computational improvements.

## 1 INTRODUCTION

Paths and cycles are fundamental structures in graph theory, playing a crucial role in various fields such as network analysis (Wang et al., 2023), chemistry (Ishida et al., 2021), computer science (AbuSalim et al., 2020), biology (Bortner and Meshkat, 2022), and social sciences (Boccaletti et al., 2023). Efficiently counting paths and cycles of varying lengths is essential for understanding graph connectivity and network redundancies, and is the foundation of many graph processing algorithms, including graph learning algorithms such as some recent Graph Neural Networks (GNN) (Bouritsas et al., 2022; Michel et al., 2023). In particular, Bouritsas et al. (2022) demonstrated that incorporating precomputed counts of paths or cycles, either at the node level or the graph level, into the feature representation can significantly enhance the expressive power of GNNs.

This problem of counting paths and cycles has been extensively studied in the literature (Harary and Manvel, 1971; Alon et al., 1997; Jokić and Van Mieghem, 2022). Among existing approaches, matrix-based formulae such as those proposed in Voropaev and Perepechko (2012) (see equation (1) for an example) are known to be the most efficient methods for paths and cycles of lengths up to six and seven, respectively (Giscard et al., 2019). This raises a significant open question: Can a deep learning algorithm discover more efficient formulae for counting paths?

In Fürer (2017) and Arvind et al. (2019), the length limits mentioned above are theoretically explained by examining the relationship between the subgraph counting problem and the $k$th-order Weisfeiler-Leman test ($k$-WL). These papers conclude that 3-WL cannot count cycles longer than seven. Concurrently, Geerts (2020) explored the connection between 3-WL and a fragment of the matrix language MATLANG (Brijder et al., 2019), defined by the operations $\mathcal{L}_3 := \{\cdot, ^{\mathbf{T}}, \text{diag}, \mathbb{1}, \odot\}$. This paper demonstrates that this fragment, when applied to adjacency matrices, distinguishes the same graph pairs as 3-WL. In order to build a 3-WL GNN, Piquenot et al. (2024) introduced a 3-WL Context-Free Grammar (CFG). We observed that, with minor modifications, this generative framework is capable of producing all the formulae previously identified by

Voropaev and Perepechko (2012). Taken together, these recent works enable to transform the search for a path/cycle counting algorithm into a CFG-constrained language generation problem where the aim is to build efficient path counting formulae.

The use of context-free grammars (CFGs) for search has gained considerable attention in the literature. In robotics, Zhao et al. (2020) demonstrated that CFGs can effectively describe robotic structures, enabling the design and adaptation of robots for diverse tasks to be formulated as a search problem within a CFG. Similarly, in the field of AutoML, the application of CFG-based search has been explored (Klinghoffer et al., 2023; Vázquez et al., 2023). In particular, Vazquez et al. (2022) proposed a grammar specifically designed for an AutoML task, where searching within this grammar enables the discovery of the optimal model for a given task.

Searching for formulae within a CFG corresponds to solving a combinatorial optimization problem of possibly infinite size. In recent years, Deep Reinforcement Learning (DRL) approaches have been proposed to address such problems (Vinyals et al., 2015; Khalil et al., 2017; Silver et al., 2018; Hubert et al., 2021; Darvariu et al., 2024). A recent success of DRL has been the discovery of more efficient matrix multiplication algorithms through a Monte Carlo Tree Search (MCTS)-based approach (Fawzi et al., 2022). MCTS-based RL algorithms typically consist of two phases: an acting phase and a learning phase. During the acting phase, the agent selects actions based on a heuristic that combines MCTS exploration with a deep neural network that predicts both policy and value function to guide the tree search. The network is then updated during the learning phase to reflect search trees from multiple iterations. As the objective is to discover efficient formulae and considering that MCTS aligns with CFG sentence generation process due to its tree-based structure, such a Deep MCTS approach is particularly well-suited to searching formulae within CFGs.

In this paper, we propose Grammar Reinforcement Learning (GRL), a deep MCTS model capable of discovering new efficient formulae for path/cycle counting within a CFG. This particular context raises a new research question: How to approximate a policy and a value function through a deep neural network within a CFG?

To address this question, we propose the Gramformer model, a transformer architecture that models Pushdown Automata (PDA), which are equivalent to CFGs. Gramformer is used to learn the policy and value functions within GRL.

When applied to the path/cycle counting problem, GRL not only recovers the formulae from Voropaev and Perepechko (2012) but also discovers new ones, whose computational efficiency is improved by a factor of two to six.

The key contributions of this paper are as follows: (i) We propose GRL, a generic DRL algorithm designed to explore and search within a given grammar. (ii) We introduce Gramformer a new transformer training pipeline compliant with the CFG/PDA framework. (iii) We propose novel state of the art explicit formulae for path/cycle counting in graphs, leading to substantial improvements in computational efficiency.

The structure of this paper is as follows: Section 2 provides the background about path/cycle counting, CFGs and PDAs, defining essential concepts. Section 3 describes the design of GRL, taking as root a given CFG. Section 4 presents Gramformer, connecting transformers and CFGs. Section 5 discusses the results of GRL on path/cycle counting tasks. Finally, we conclude with a summary of our contributions and suggest avenues for future research.

## 2 BACKGROUND

### 2.1 PATH AND CYCLE COUNTING

Path and cycle counting in graphs can be performed at multiple levels: graph, node, and edge. At the **graph level**, all possible paths or cycles of a given length within the graph are counted. At the **node level**, the focus is on counting paths starting at a specific node, as well as cycles that include the node. At the **edge level**, for any non-negative integer $l$, let $P_l$ represent the $l$-path matrix where $(P_l)_{i,j}$ is the number of $l$-length paths connecting vertex $i$ to vertex $j$. Additionally, for $l > 2$, let $C_l$ represent the $l$-cycle matrix where $(C_l)_{i,j}$ indicates the number of $l$-cycles that include vertex $i$ and its adjacent vertex $j$.

As mentioned in Section 1, path/cycle counting has been extensively tackled in the literature. In the early 1970s, Harary and Manvel (1971) introduced algorithms for counting cycles up to length five at the graph level. Two decades later, Alon et al. (1997) refined these algorithms, extending cycle counting to lengths of up to seven, and conjectured that these methods can also be adapted to count cycles at the node level. Later Voropaev and Perepechko (2012) established a relationship between the counting of $l$-cycles at the edge level and the counting of $(l-1)$-paths at the edge level using a simple formula. By deriving explicit formulae for the counting of paths of length up to six at the edge level, they were able to compute the number of cycles of length up to seven. More recently, Jokić and Van Mieghem (2022) rediscovered the formulae for paths of length up to four from Voropaev and Perepechko (2012). In contrast, Giscard et al. (2019) proposed an algorithm capable of counting cycles and paths of arbitrary lengths. However, they acknowledged that their method is slower than those presented by Alon et al. (1997) and Voropaev and Perepechko (2012). Specifically, since the latter algorithms are based on matrix multiplication, they exhibit a computational complexity of $O\left(n^3\right)$, where $n$ is the number of nodes. As noted by Giscard et al. (2019), these matrix-based approaches remain the most efficient known methods for counting paths and cycles of lengths up to six and seven, respectively.

## 2.2 CONTEXT-FREE GRAMMAR.

Throughout this paper, we employ standard formal language notation: $\Gamma^*$ denotes the set of all finite-length strings over the alphabet $\Gamma$, and $\varepsilon$ represents the empty string. The relevant definitions used in this context are as follows:

**Definition 2.1** (Context-Free Grammar)
A Context-Free Grammar (CFG) $G$ is defined as a 4-tuple $(V, \Sigma, R, S)$, where $V$ is a finite set of variables, $\Sigma$ is a finite set of terminal symbols, $R$ is a finite set of production rules of the form $V \to (V \cup \Sigma)^*$, and $S$ is the start variable. *Note that $R$ fully characterizes the CFG, following the convention that the start variable is placed on the top left and that the symbol $|$ represents "or".*

**Definition 2.2** (Derivation)
Let $G$ be a CFG. For $u, v \in (V \cup \Sigma)^*$, we define $u \implies v$ if $u$ can be transformed into $v$ by applying a single production rule, and $u \stackrel{*}{\implies} v$ if $u$ can be transformed into $v$ by applying a sequence of production rules from $G$.

**Definition 2.3** (Context-Free Language)
A set $B$ is called a Context-Free Language (CFL) if there exists a CFG $G$ such that $B = L(G) := \{w \mid w \in \Sigma^* \text{ and } S \stackrel{*}{\implies} w\}$.

The generation process in a CFG involves iteratively replacing variables with one of their corresponding production rules, starting from the start variable, until only terminal symbols remain.

As mentioned in Section 1, it is well known that CFGs are equivalent to PDAs (Schneider, 1968; Caucal, 1995; Baeten et al., 2023; DuSell and Chiang, 2024). Usually, PDA are language acceptor, but by relabelling the input as output of the PDA, the same PDA can be used as language generator. Thus the following subsection is dedicated to defining PDA.

## 2.3 PUSHDOWN AUTOMATON

**Definition 2.4** (PushDown Automaton)
A PushDown Automaton (PDA) is defined as a 7-tuple $P = (\mathcal{Q}, \Sigma, \Gamma, \delta, q_0, Z, F)$ where $\mathcal{Q}$ is a finite set of states, $\Sigma$ is a finite set of symbols called the input alphabet, $\Gamma$ is a finite set of symbol called the stack alphabet, $\delta$ is a finite subset of $\mathcal{Q} \times (\Sigma \cup \{\varepsilon\}) \times \Gamma \to \mathcal{Q} \times \Gamma^*$, the transition function, $q_0 \in \mathcal{Q}$ is the start state, $Z \in \Gamma$ is the initial stack symbol, $F \subseteq \mathcal{Q}$ is the set of accepting states.

In the case of PDA corresponding to CFG, the input alphabet $\Sigma$ corresponds to the terminal symbol alphabet. The stack alphabet $\Gamma$ consits of $V \cup \Sigma$, which is the union of the set of variables (non-terminal symbols) and the terminal symbols. For such a PDA, there are only two states: $q_0$, the initial state and, $q_1 \in F$, the accepting state. The initial stack symbol is $Z = S$, where $S$ is the start variable of the CFG. The transition function $\delta$ consists of two types of transitions:

- **Transcription transitions**: If the top of the stack is a terminal symbol $a \in \Sigma$, the transition is of the form $\delta(q_0, a, a) = \{(q_0, \varepsilon)\}$. This indicates that the system remains in state $q_0$, outputs the symbol $a$, and removes $a$ from the stack.

- **Transposition transitions**: If the top of the stack is a variable $\nu \in V$, the transition is of the form $\delta(q_0, \varepsilon, \nu) = \{(q_0, r), r \in V_\nu\}$, where $V_\nu \subset R$ is the subset of rules for $\nu$. This means that the system stays in state $q_0$, produces no output, and replaces $\nu$ with the rule $r$ on the stack.

In the same way that production rules fully defines a CFG, the transition function $\delta$ completely specifies a PDA. For a PDA constructed from a CFG, the transposition transitions alone are sufficient to define the automaton.

A PDA generates a string by starting in the initial state $q_0$, with the stack initialized to $Z$ and the generated string $s$ initialized to $\varepsilon$. The PDA then processes the top symbol $t$ of the stack according to the transition function $\delta$. If $t \in \Sigma$, a transcription occurs: $t$ is popped from the stack and appended to the output string $s$. If $t \in V = \Gamma \setminus \Sigma$, a transposition occurs: $t$ is popped from the stack, and some $v \in \{v, (q_0, v) \in \delta(q_0, \varepsilon, t)\}$ is pushed onto the stack. Since $v \in \Gamma^*$, it may consist of multiple symbols, which are pushed onto the stack in reverse order. The process continues until the stack is empty, at which point the PDA transitions to the accepting state $q_1$, and the generated string $s$ is a member of the language of the corresponding CFG.

## 3 GENERATING PATH/CYCLE COUNTING FORMULA THROUGH GRL

The following subsection presents a specific CFG (see Section 2) designed to address the open problem of path counting.

### 3.1 FROM PATH MATRIX FORMULAE TO THE CFG $G_3$

Let $\mathcal{G} = (\mathcal{V}, \mathcal{E})$ denote an undirected graph, where $\mathcal{V} = [\![1, n]\!]$ represents the set of $n$ nodes, and $\mathcal{E} \subseteq \mathcal{V} \times \mathcal{V}$ represents the set of edges. We define the adjacency matrix $A \in \{0, 1\}^{n \times n}$, that encodes the connectivity of $\mathcal{G}$, the identity matrix $\mathrm{I} \in \{0, 1\}^{n \times n}$, and the matrix $\mathrm{J} \in \{0, 1\}^{n \times n}$, that is filled with ones except along the diagonal.

In the work of Voropaev and Perepechko (2012), all of the proposed formulae are linear combinations of terms composed of matrix multiplications and Hadamard products (denoted by $\odot$) applied exclusively on the arguments $A$, I, J. For example, the matrix formula in equation (1) is used to count the number of 3-paths between two nodes in the graph. The formulae for path of length four to six can be found in Appendix D of the supplementary material.

$$P_3 = \mathrm{J} \odot A^3 - (\mathrm{I} \odot A^2)A - A(\mathrm{I} \odot A^2) + A \tag{1}$$

To generate the terms of Voropaev's formulae, we define the CFG $G_3$ in equation (2).

$$M \to (M \odot M) \,|\, (MM) \,|\, A \,|\, \mathrm{I} \,|\, \mathrm{J}. \tag{2}$$

Voropaev's formulae are linear combinations of sentences of $L(G_3)$. This ensures that the problem of counting paths of length up to 6 can be addressed through finding linear combinations of sentences of $L(G_3)$.

Additionally, we prove in Appendix A that $G_3$ is 3-WL equivalent, resulting in Theorem 3.1.

**Theorem 3.1** (3-WL CFG)
*$G_3$ is as expressive as 3-WL*

While CFGs are theoretical objects, PDAs are the practical tools for processing and applying the production rules of a CFG to ensure the correct generation of valid sentences according to the grammatical structure. The following subsection derives a PDA (see Section 2) from $G_3$.

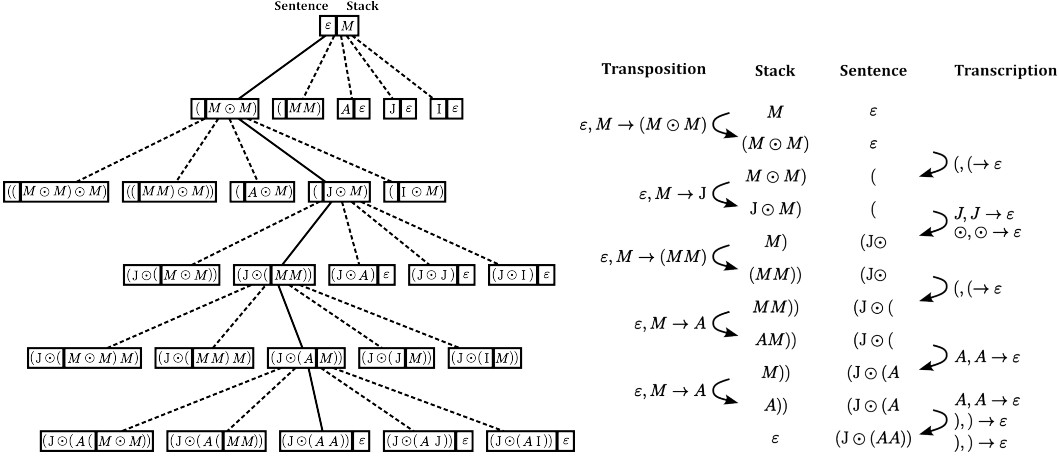

Figure 1: The left diagram illustrates a path in the derivation tree of the PDA $D_3$ which generates the sentence $J \odot A^2 \in L(G_3)$. The right diagram details the process of generating this sentence, emphasizing the transcription and transposition loops. As depicted, the stack fills during transposition steps and empties during transcription steps, eventually leading to the derivation of a sentence from the language.

## 3.2 FROM $G_3$ TO THE PDA $D_3$

We denote as $D_3$ the PDA described by the following transition $\delta$:

$$\delta(q_0, \varepsilon, M) = \{(q_0, (M \odot M)), (q_0, (MM)), (q_0, A), (q_0, \mathrm{I}), (q_0, \mathrm{J})\},$$

which corresponds directly to the production rules of $G_3$.

Figure 1 illustrates how the sentence $(\mathrm{J} \odot (AA)) = \mathrm{J} \odot A^2 \in L(G_3)$ is generated by the PDA $D_3$.

## 3.3 SEARCH IN $D_3$ THROUGH GRAMMAR REINFORCEMENT LEARNING

To find efficient formulae for path and cycle counting, we propose a two step strategy as illustrated by Figure 2. The first step is to generate a set of sentences belonging to $G_3$ by the $D_3$ generation process. The second step compares a linear combination of this set with a ground truth matrix in order to evaluate the corresponding formula. In the following of this subsection, we detail each of these steps.

As stated before, the tree structure of a sentence generation within PDA (see Figure 2) aligns with MCTS algorithm. Such algorithms have been proposed and refined over the last decade to guide the search within trees with a general heuristic (Świechowski et al., 2023). In this work, we propose an MCTS-based DRL algorithm, termed Grammar Reinforcement Learning (GRL) adapted to the path counting open problem, generating sets of different sentences.

In GRL, MCTS performs a series of walks through the PDA, which are stored in a search tree. The nodes of the tree represent states $I$, which are a concatenation of the written terminal symbols and the stack. The edges correspond to actions defined by the CFG rules $r$ that can be applied at those states.

Each walk begins at the start state $I_0 = \{Z, \cdots, Z\}$, whose cardinality is the number of desired sentences, and terminates when a state contains only terminal symbols. Such terminal states are sets of sentences located in leaf nodes. For each state-action pair $(I, r)$, the algorithm tracks the visit count $N(I, r)$, the empirical rule value $Q(I, r)$, and two scalars predicted by a neural network: a policy probability $\pi(I, r)$ and a value $v(I, r)$. From a reinforcement learning perspective, $\frac{Q(I,r)}{N(I,r)}$ represents the expected return over all possible trajectories that originate from state $I$ and follow action $r$. At each intermediate state, a rule action $r$ is selected according to the following equation:

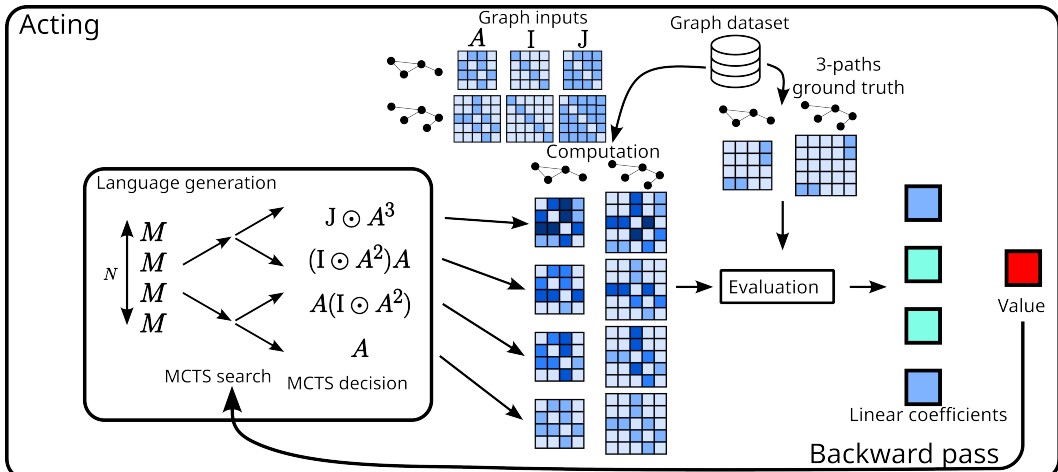

Figure 2: From left to right: The agent selects a set of $N$ sentences based on an MCTS heuristic. These sentences are computed for a given set of graphs. The computation is then evaluated against a ground truth, yielding a linear combination of the sentences and a value representing their pertinence. This value is subsequently backpropagated through the MCTS search tree.

$$\operatorname*{argmax}_{r} \quad \alpha Q(I, r) + (1 - \alpha)v(I, r) + c(I)\pi(I, r)\frac{\sqrt{\sum_a N(I, a)}}{1 + N(I, r)}, \tag{3}$$

where the exploration factor $c(I)$ regulates the influence of the policy $\pi(I, r)$ relative to the Q-values, adjusting this balance based on the frequency of node traversal. The parameter $\alpha \in [0, 1]$ controls the reliance on neural network predictions. After a walk reaches a leaf node, the visit counts and the values are updated via a backward pass.

To update the values, it is necessary to evaluate a leaf node. Its associated set of sentences is computed for a collection of graphs, and a linear combination of these computed sentences is derived by comparing them against a ground truth for each graph. The resulting value, which reflects the relevance of the sentences to a specific path counting problem, is used to empirically update the tree that is constructed during sentence generation. The derivation of this linear combination is detailed in Appendix B, with a specific focus on Figure 8. A concrete example of this approach, applied to the problem of counting 3-paths within $G_3$, is shown in Figure 2. To encourage the generation of efficient sentences, each CFG rule $r$ is penalized by a value $P_r$ in the reward definition, reflecting its computational cost. Additionally, to prevent the generation of overly long sentences, the number of characters is constrained by a maximum limit, $C_{\max}$.

After a sufficient number of MCTS, the sequences of nodes and edges from each walk are used to train the neural network. The ratio $N(I, r)/N(I)$ provides a policy derived from MCTS exploration, while $Q(I, r)$ represents the empirical expected return for the current state. The policy is learned using a Kullback–Leibler (KL) divergence loss, and the value function is trained using a mean squared error (MSE) loss. The pseudo-code for each algorithm of GRL is provided in Appendix F. Figure 3 depicts both the acting and the learning parts.

The neural network—serving as a memory of the search trees that the agent has previously explored—must be capable of learning the policy and value distributions of the sentence generation within the CFG/PDA. As discussed in Section 1, designing an architecture that can effectively learn within a CFG/PDA remains an open research question. In the next section, we present the neural network used for estimating the policy and value functions in the GRL algorithm.

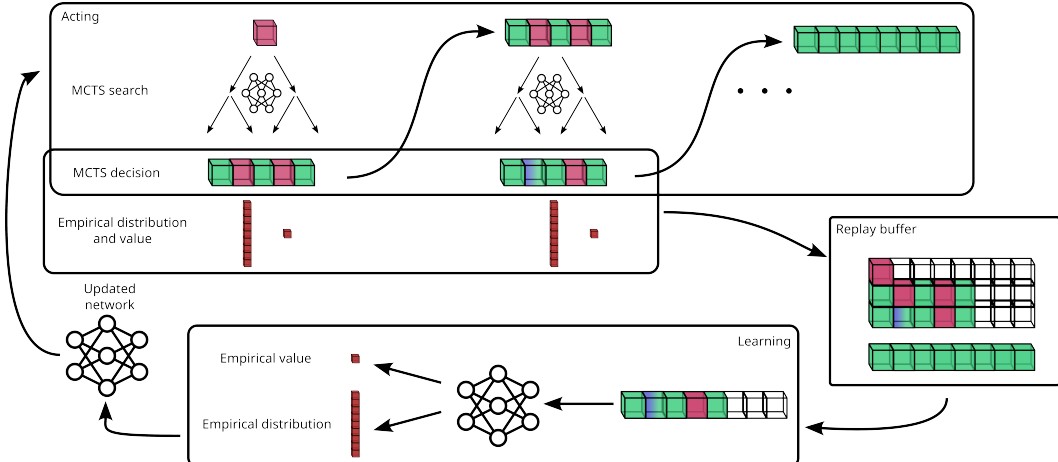

Figure 3: In the acting phase, rules are selected based on both the MCTS algorithm and the neural network outputs. Each time MCTS selects a node, the decision, empirical policy, and value of the node are stored in a replay buffer. During the learning phase, the neural network is updated by predicting the policy and value functions based on the decisions stored in the replay buffer.

# 4  GRAMFORMER

Since our problem is related to the generation of sentences within a language, a transformer architecture fits with this CFG framework. Central to this architecture is the concept of tokens, which represent individual units of input data.

We propose Gramformer, a transformer architecture that follows the production rules of a given CFG, through a PDA. It relies on the assignment of the elements of the transition function $\delta$ into three distinct sets of tokens. Recall that $\delta$ can be partitioned into two subsets: the transcription set $\delta_w$ (for writing) and the transposition set $\delta_r$ (for replacing). Specifically, $\delta_r = \{\delta(q_0, \varepsilon, \nu) = \{(q_0, v), v \in R_\nu)\}, \nu \in V\}$, where each $\nu \in V$ represents a variable in the CFG.

For each variable $\nu \in V$, we define a **variable token** corresponding to $\nu$. For each element in $\delta(q_0, \varepsilon, \nu) \in \delta_r$, we define a **rule token** representing the specific production rule. This rule token is divided in two subsets. If the rule contains a variable, the token is classified as a **non-terminal rule token**. If the rule consists only of terminal symbols (i.e., $v \in \Sigma$), the token is classified as a **terminal rule token**. Any symbol in $\delta_w$ is assigned as part of the **terminal token** set along with the terminal rule token.

For each variable token, a corresponding mask is provided. This mask indicates the rule tokens associated with that variable. Figure 4 illustrates this framework applied to the CFG $G_3$ that contains only one variable. An example of a CFG with more variables is provided in Figure 9 of Appendix C. Once all tokens have been defined, the Gramformer is tasked with predicting two ouputs for a given input state. The first output is the probability of selecting the production rule for a given variable. The second one is a scalar that corresponds to the value of the given state.

Gramformer follows a classical encoder-decoder architecture with self-attention and cross-attention mechanisms. At any time, the model's input $I$ consists of the concatenation of the stack and the set of terminal symbols generated so far, representing a state.

The input $I$ is read until a variable token associated to a variable symbol is encountered. This token, denoted as $\nu$ is passed to the encoder. The decoder receives the encoder output and the input $I$. The first output of the decoder is combined with the mask corresponding to $\nu$, so that tokens not associated with $\nu$ are set to $-\infty$. This masking ensures that, when the softmax function is applied to the first decoder's output, it yields a valid probability distribution over the rules of $\nu$.

The pseudo-code for each algorithm in this framework is provided in Appendix F. Figure 5 depicts the Gramformer process for a given input.

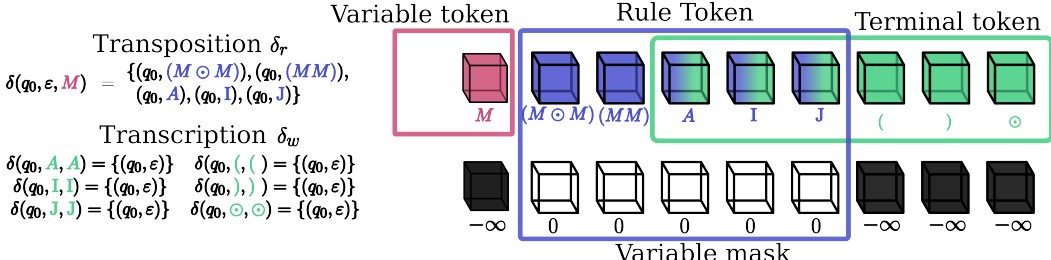

Figure 4: From PDA to grammar tokens: $D_3$ is turned into three sets of tokens. The corresponding variables of each element of $\delta_r$ are turned into variable tokens. For each variable token, a set of rule tokens is defined. Eventually, for every corresponding terminal symbols of $\delta_w$ a terminal token is defined. In the end, for each variable token, a variable mask is defined.

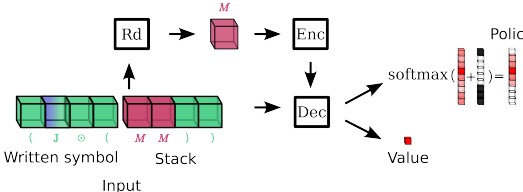

Figure 5: The input is read until the first variable token (**Rd**). This token is passed to the encoder (**Enc**). The decoder (**Dec**) receives the encoder output and the input. The first output of the decoder is combined with the mask corresponding to variable token to generate a policy. The second output is the value.

Note that Gramformer, in an autoregressive mode of operation, can generate sentences within a CFG, simulating a PDA. Figure 6 and Figure 10 of the Appendix C illustrate this generation of the sentence $(\mathrm{J} \odot (AA)) \in L(G_3)$ using the Gramformer architecture coupled with a replace block. The path in the derivation tree of $D_3$ resulting in the generation of this sentence is provided in Figure 1.

We now have the necessary components to use GRL on the path counting problem, which is described in the following section.

## 5 FINDING MORE EFFICIENT FORMULAE FOR COUNTING WITH RL.

To address the problem of path counting at the edge level, we apply GRL using a slightly modified version of the grammar $G_3$, denoted $\tilde{G}_3$. This grammar generates matrices of $L(G_3)$ with a null diagonal, reducing the search space. For more details on this modified grammar, please refer to Appendix C.

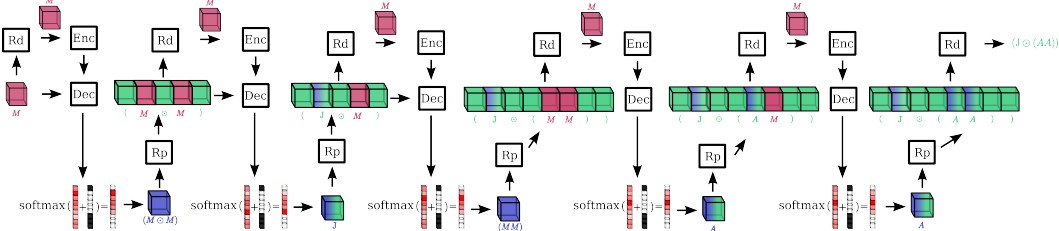

Figure 6: A sequence of tokens is fed into the transformer architecture, beginning with the start variable token. The transformer ouputs the rule token corresponding to the current variable token. The predicted rule's corresponding variables and terminal symbols replace the current variable token, producing a new sequence of tokens. The process is repeated, until no variables remains. At this last step, a sentence from the grammar is generated.

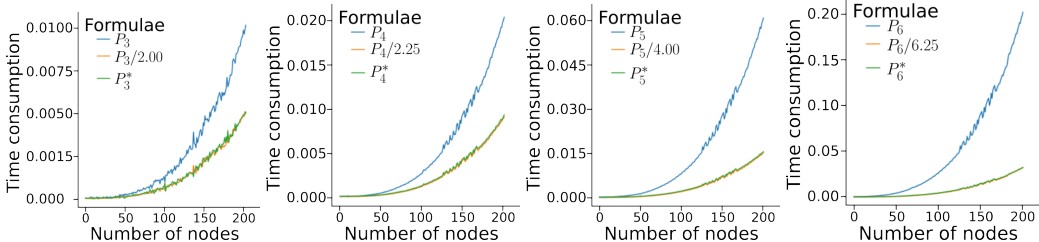

Figure 7: Comparison of the time consumption of $P_l$ and $P_l^*$ in function of the number of nodes for $l \in \{3, 4, 5, 6\}$. Each time, the yellow correspond to the time computation of $P_l$ divided by the theoretical gain of time consumption.

The primary objective of this experiment was to demonstrate that GRL can successfully derive the path counting formulae $P_l$ proposed in Voropaev and Perepechko (2012). Specifically, for $l = 2$, GRL successfully identified the formula $P_2$. For path lengths $l \in 3, 4, 5, 6$, GRL not only derived the $P_l$ formulae but also discovered more efficient alternatives, denoted as $P_l^*$. These new formulae significantly reduce the time complexity of $l$-path counting by factors of 2, 2.25, 4, and 6.25, respectively. The formulae for $P_2^*$ through $P_4^*$ are provided below, while those for $P_5^*$ and $P_6^*$ can be found in Appendix D.

$$
\begin{aligned}
P_2^* &= \mathrm{J} \odot A^2, \\
P_3^* &= \mathrm{J} \odot (A(\mathrm{J} \odot A^2)) - A \odot (A\mathrm{J}), \\
P_4^* &= \mathrm{J} \odot (A(\mathrm{J} \odot (A(\mathrm{J} \odot A^2)))) - \mathrm{J} \odot (A(A \odot (A\mathrm{J}))) \\
&\quad - \mathrm{J} \odot ((A \odot (A\mathrm{J}))A) - A \odot ((A \odot A^2)\mathrm{J}) + 2A \odot A^2.
\end{aligned}
$$

For each formula, we prove in Appendix D that $P_l = P_l^*$, leading to the following theorem.

**Theorem 5.1** (Efficient path counting)
*For $l \in \{2, 3, 4, 5, 6\}$, $(P_l^*)_{i,j}$ computes the number of $l$-paths starting at node $i$ and ending at node $j$.*

It is visually obvious that $P_3^*$ is more compact than $P_3$. To quantify this, we compare the number of matrix multiplications required, which allows us to derive the ratio of time complexity between the formulae. The theoretical time savings between $P_l$ and $P_l^*$ are detailed in Appendix D.

We also assessed the empirical time savings across various random graphs. For each graph, the time required to compute each formula was recorded, and the average computation time was calculated for graphs of the same size. To compare these results with the theoretical time savings, we divided the mean computation time of $P_l$ by the corresponding theoretical time reduction factor. The results of these experiments are presented in Figure 7, demonstrating a strong alignment between empirical and theoretical gains. This confirms the significant time savings provided by the new formulae discovered by GRL and supports our theoretical analysis.

In Appendix D, we derive the cycle-counting formulae based on the work of Voropaev and Perepechko (2012), using the relation $C_{l+1} = A \odot P_l$. Additionally, we provide a detailed explanation of how $P_l^*$ counts $l$-path establishing a new methodology for deriving formulae.

Since 3-WL cannot count the 7-paths (Fürer, 2017), theorem 3.1 leads to the incapability for $G_3$ to count it either. To go beyond the 6-paths counting, a more expressive grammar is needed.

In Appendix E, we evaluate GRL on directed graphs and compare its performance to baseline algorithms. GRL stands out as the only approach capable of discovering novel matrix formulae for counting paths of lengths 4 to 5 in directed graphs.

## 6 CONCLUSION

This paper introduces Gramformer, a deep learning architecture that learns a policy and a value function within a CFG/PDA framework, by assigning tokens to elements of the transition function of a PDA. Used within the GRL algorithm, it effectively addresses the question "Can a deep learning algorithm discover efficient set of sentences for a given task".

Instantiated over the grammar $G_3$ to solve the path counting problem, GRL provides efficient formulae that are linear combinations of sets of sentences in $L(G_3)$. These formulae of enhanced computational efficiency by factors ranging from 2 to 6.25 demonstrate the ability of GRL to not only discover explicit formulae for counting paths, but also to provide new ways of designing such formulae.

For paths longer than 6, future research should aim to characterize $k$-WL CFGs to bypass the theoretical limit on path counting of $G_3$. Such a characterization will enable the application of GRL to uncover more explicit formulae for substructure counting across graph structures.

Moreover, applying GRL to real-world datasets to derive formulae for various tasks represents a promising direction for future exploration as the grammar provides a link to substructures and thus interpretability.

This approach could potentially improve the applicability and effectiveness of GRL in practical scenarios, thereby broadening its impact.

## ACKNOLEDGEMENTS

The authors acknowledge the support of the French Agence Nationale de la Recherche (ANR) under grant ANR-21-CE23-0025 (CoDeGNN project) and grant ANR-20-LCV1-0009 (Labcom Lisa). This work was supported by computational resources provided by CRIANN (Centre Régional Informatique et d'Applications Numériques de Normandie).

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

This document provides additional content to the main paper.

## A  CFGs AND PDAs

This section provides the proof of theorem 3.1 of Section 2 and more details about PDA.

Even if $G_3$ is different from the 3-WL CFG proposed in Piquenot et al. (2024), they share the same expressive power. Indeed, in the context of this paper, we are not limited by the depth of the CFG while the goal of the grammar reduction in Piquenot et al. (2024) was to keep the expressiveness of the CFG at a given depth.

It is important to note that in a separative point of view, we separate graphs with scalar, a CFG $G$ separates two graphs $\mathcal{G}_1$ and $\mathcal{G}_2$ if there exists a sentence $s \in L(G)$ such that $s(A_{\mathcal{G}_1}) \neq s(A_{\mathcal{G}_2})$. Knowing that, we have the following proposition and theorem relative to the expressive power of $G_3$.

**Proposition A.1**
*Assume we have a sentence $s$ that is the sum of two sentences $s_1$ and $s_2$. If $s$ separates $\mathcal{G}_1$ and $\mathcal{G}_2$, then it is necessary that $s_1$ or $s_2$ separate $\mathcal{G}_1$ and $\mathcal{G}_2$.*

*Proof.* Assume for the sake of contradiction that neither $s_1$ nor $s_2$ can separate $\mathcal{G}_1$ and $\mathcal{G}_2$. Then

$$s(A_{\mathcal{G}_1}) = s_1(A_{\mathcal{G}_1}) + s_2(A_{\mathcal{G}_1})$$
$$= s_1(A_{\mathcal{G}_2}) + s_2(A_{\mathcal{G}_2}) = s(A_{\mathcal{G}_2}).$$

That is absurd. □

**Theorem A.1** (3-WL CFG)
*$G_3$ is as expressive as 3-WL*

*Proof.*

$$V_c \to M V_c \mid \mathbb{1} \tag{4}$$
$$M \to (M \odot M) \mid (MM) \mid \operatorname{diag}(V_c) \mid A.$$

We will start from the CFG (4) that was proven to be 3-WL equivalent. We show that $V_c$ variable and $\operatorname{diag}(V_c)$ can be removed.

First of all, we have that for any matrix $N$ and vector $w$, $Nw = (N \odot \mathrm{I})w + (N \odot \mathrm{J})w$, since a sentence in the CFG (4) consists on a sum other the resulting vector, we have with the help of proposition A.1 that vectors $(N \odot \mathrm{I})w$ and $(N \odot \mathrm{J})w$ have a better separability than $Nw$. To remove $V_c$ variable, we first have $\mathrm{I} = \operatorname{diag}(\mathbb{1})$. Then for any matrix $N$ and vector $w$, we have that $(N\operatorname{diag}(w)\,\mathrm{J}) \odot \mathrm{I} = \operatorname{diag}((N \odot \mathrm{J})w)$ and $(N\operatorname{diag}(w)) \odot \mathrm{I} = \operatorname{diag}((N \odot \mathrm{I})w)$.

$$(N\operatorname{diag}(w)\,\mathrm{J})_{i,i} = \sum_{l,m} N_{i,l}\operatorname{diag}(w)_{l,m}\,\mathrm{J}_{m,i}$$
$$= \sum_l N_{i,l} w_l \mathrm{J}_{l_i}$$
$$= \sum_l N_{i,l} \mathrm{J}_{i_l} w_l$$
$$= \sum_l (N \odot \mathrm{J})_{i,l} w_l = ((N \odot \mathrm{J})w)_i,$$

$$(N\operatorname{diag}(w))_{i,i} = \sum_l N_{i,l}\operatorname{diag}(w)_{l,i}$$
$$= N_{i,i} w_i$$
$$= \sum_l (N \odot \mathrm{I})_{i,l} w_l = ((N \odot \mathrm{I})w)_i$$

The conclusion can be made by induction. We obtain $G_3$ as expressive as 3-WL. □

To give more insight in the construction of PDA from CFG, consider the PDA $D_3$, which corresponds to the CFG $G_3$:

$$D_3 = (\{q_0, q_1\}, \{A, \mathrm{I}, \mathrm{J}, (,), \odot\}, \{M, A, \mathrm{I}, \mathrm{J}, (,), \odot\}, \delta, q_0, M, q_1).$$

where the transition relation $\delta$ is defined as follows:

$$\delta(q_0, \varepsilon, M) = \{(q_0, (M \odot M)), (q_0, (MM)), (q_0, A), (q_0, \mathrm{I}), (q_0, \mathrm{J})\}$$
$$\delta(q_0, A, A) = \delta(q_0, \mathrm{I}, \mathrm{I}) = \delta(q_0, \mathrm{J}, \mathrm{J}) = \delta(q_0, (, () = \delta(q_0, ), )) = \delta(q_0, \odot, \odot) = \{(q_0, \varepsilon)\}$$
$$\delta(q_0, \varepsilon, \varepsilon) = \{(q_1, \varepsilon)\}.$$

In the same way that production rules fully defines a CFG, the transition relation $\delta$ completely specifies a PDA. For a PDA constructed from a CFG, the transposition transitions alone are sufficient to define the automaton. For instance, $D_3$ can be fully described by the transition:

$$\delta(q_0, \varepsilon, M) = \{(q_0, (M \odot M)), (q_0, (MM)), (q_0, A), (q_0, \mathrm{I}), (q_0, \mathrm{J})\},$$

which corresponds directly to the production rules of $G_3$.

## B    ON THE EVALUATION OF GRL IN THE CONTEXT OF PATH COUNTING

We remind the acting phase of GRL described in section 4. In GRL, an agent generates a set of sentences, $\mathcal{S}$, using a pushdown automaton corresponding to a given CFG. For a given set of graphs, the agent computes the results of each sentence in $s$. A linear combination of these computed results is then derived and compared to the ground truth path counts, which yields a reward $R_s$. This section aims to detailed this evaluation process in the case of GRL applied to $G_3$.

In the case of $G_3$, the set of computed sentences for a given set of graphs results into a set of matrices for each sentence. We have then $s$ sets of $g$ matrices, where $s$ is the number of sentences and $g$ the number of graphs. Along with this, we have a set of $g$ matrices of ground truth. For the sake of explanation, we assume that each graphs have the same size $n$. Then we chose $s$ indices $\iota_1, \cdots, \iota_s$ and a graph $\mathcal{G}$ such that, the matrix $E$ of size $s \times s$, where $E_{i,j} = s_j(A_{\mathcal{G}})_{\iota_i}$, is invertible. If such a matrix does not exist, we penalise the set of sentences by attributing a negative value. Along with the construction of $E$, we define the vector $v$ with the ground truth matrix of $\mathcal{G}$, $T_{\mathcal{G}}$ by $v_i = (T_{\mathcal{G}})_{\iota_i}$.

The linear combination is then obtained by resolving the equation $Ex = v$. Then, the linear combination $\sum_i x_i s_i(A_{\mathcal{G}})$ is compared to the ground truth $T_{\mathcal{G}}$ for all graphs $\mathcal{G}$ resulting in the value $r_{\mathcal{S}}$. This value encompasses the pertinence of the set of sentence $\mathcal{S}$ over a specific path counting problem. Figure 8 depicts this evaluation procedure.

The existence of Voropaev and Perepechko (2012) formulas, ensure that there exist a linear combination of sentences that addresses the path counting of length up to 6.

## C    A CFG TO COUNT AT EDGE LEVEL

In our investigation of substructure counting at the edge level for the grammar $G_3$, we focus on the non-diagonal elements of the involved matrices. To streamline this process, we introduce an alternative context-free grammar, denoted as $\tilde{G}_3$, which is equally expressive as $G_3$ but specifically tailored for edge-related computations. The grammar is defined as follows:

$$\begin{aligned} E &\to (E \odot M) \,|\, (NE) \,|\, (EN) \,|\, A \,|\, \mathrm{J} \\ N &\to (N \odot M) \,|\, (N \odot N) \,|\, \mathrm{I} \\ M &\to (MM) \,|\, (EE) \end{aligned} \tag{5}$$

In $\tilde{G}_3$, the variable $E$ represents matrices with zero on the diagonal, corresponding to edges in the graph, while $N$ represents diagonal matrices, corresponding to nodes, and $M$ represents general matrices. The start variable is $E$ as we aim to focus on edge-level structures. The production rules for each variable describe valid operations and combinations within $G_3$ that yield matrices corresponding to that variable.

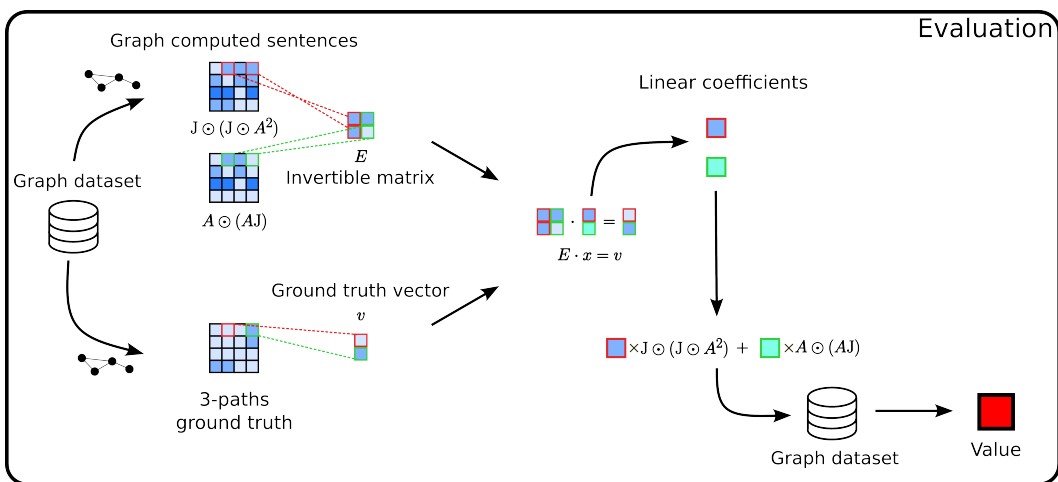

Figure 8: The evaluation process of GRL for path counting involves deriving an invertible matrix $E$ from the computed sentences corresponding to a given graph, alongside a ground truth vector $v$. The solution of the equation $Ex = v$ provides a linear combination of the computed sentence results. This linear combination is then compared to the ground truth across the entire graph dataset, yielding a value that reflects the effectiveness and relevance of the set of sentences in solving the path counting problem.

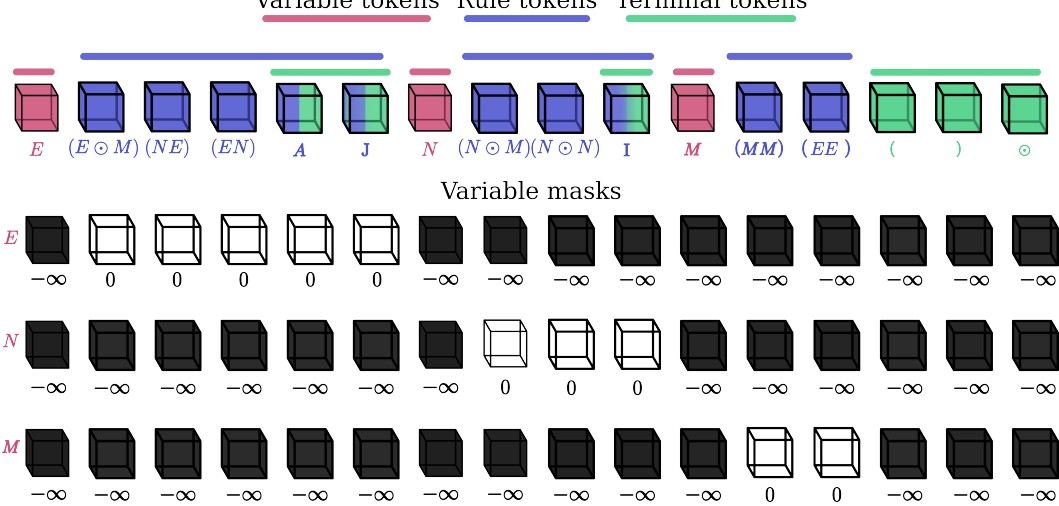

Figure 9: At the top, the tokens of the Graformer derived from $\tilde{G}_3$ are separated into three sets. Below, the variable masks of Gramformer are defined to correspond to the rule tokens of their corresponding variable token.

In the case of $N$, matrix multiplication is omitted because, for diagonal matrices, the matrix product behaves like the Hadamard product. This choice reduces computational complexity without sacrificing expressiveness.

# D  PATH AND CYCLE COUNTING

This section contains the proof of theorem 5.1 of Section 5 and provides a detailed explanation of how $P_l^*$ counts $l$-path. In the following, all graphs are assumed to be simple, i.e., they contain no self-loops. This assumption aligns with the search for paths, as self-loops cannot contribute to any

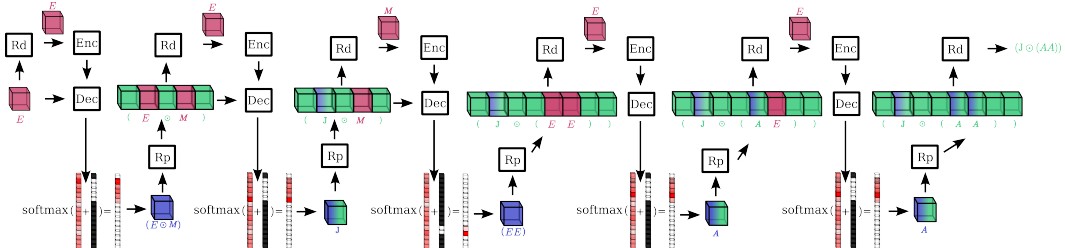

Figure 10: Generation of the sentence $J \odot A^2 \in L(\tilde{G}_3)$, following a PDA procedure guide by a Gramformer policy.

path due to the repetition of a node when traversing through a self-loop. Consequently, the adjacency matrix $A$ will always have a zero diagonal. Furthermore, this ensures that $J \odot A = A$.

As GRL found more efficient formulas to calculate paths and cycles at edge-level in a graph, we tried to prove that such formulas are correct, and by doing so, we found the following lemma that helps to reduce the computation cost.

**Lemma D.1**
*Let $N, M$ and $P$ be square matrices of the same size, such that $N_{i,i} = \sum_k M_{i,k}$ for all indices $i$. Then we have*

$$P \odot (MJ) = (I \odot N)P - P \odot M$$

*Proof.* We have

$$
\begin{aligned}
(P \odot (MJ))_{i,j} &= P_{i,j}(\sum_k M_{i,k} - M_{i,j}) \\
&= P_{i,j}N_{i,i} - P_{i,j}M_{i,j},
\end{aligned}
\tag{6}
$$

and

$$
\begin{aligned}
((I \odot N)P - P \odot M)_{i,j} &= \sum_k (I \odot N)_{i,k}P_{k,j} - P_{i,j}M_{i,j} \\
&= N_{i,i}P_{i,j} - P_{i,j}M_{i,j}.
\end{aligned}
\tag{7}
$$

From equations (6) and (7), we can conclude. $\qquad\square$

**2-paths and 3-cycles** The most effective explicit formula discovered to date for calculating the number of 2-paths connecting two nodes was proposed by Voropaev and Perepechko (2012), it is

$$P_2 = J \odot A^2. \tag{8}$$

Following the formula for the $l$-cycle proposed in Voropaev and Perepechko (2012), we obtain for the 3-cycle the following formula

$$C_3 = A \odot P_2 = A \odot A^2. \tag{9}$$

Without any surprise, our architecture found the same formulas for both 2-path and 3-cycle.

**3-paths and 4-cycles** The most effective explicit formula discovered to date for calculating the number of 3-paths connecting two nodes was proposed by Voropaev and Perepechko (2012), it is

$$P_3 = J \odot A^3 - (I \odot A^2)A - A(I \odot A^2) + A. \tag{10}$$

Following the formula for the $l$-cycle proposed in Voropaev and Perepechko (2012), we obtain for the 3-cycle the following formula

$$C_4 = A \odot P_3 = A \odot A^3 - A(I \odot A^2) - (I \odot A^2)A + A. \tag{11}$$

Obviously our architecture found $P_3$, but surprisingly, it found a more compact formula.

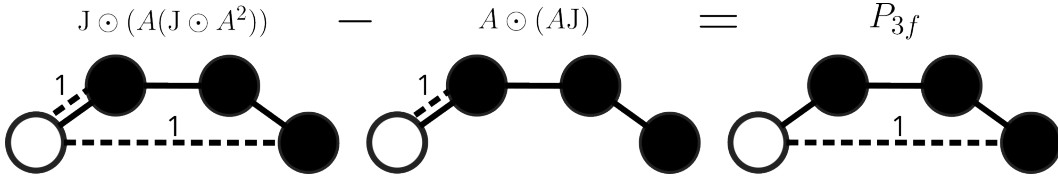

Figure 11: computation of $P_3^*$ for an example graph, the dashed lines indicate an entry in the column corresponding to the white node in the matrix associated with the term specified above.

**Theorem D.2**
*The following formula, denoted as $P_3^*$, computes the number of 3-paths linking two nodes*

$$P_3^* = \mathrm{J} \odot (A(\mathrm{J} \odot A^2)) - A \odot (A\mathrm{J}). \tag{12}$$

*Proof.* We will show that $P_3 = P_3^*$. Firstly, we have

$$\mathrm{J} \odot A^3 - A(\mathrm{I} \odot A^2) = \mathrm{J} \odot ((A(\mathrm{J} + \mathrm{I}) \odot A^2)) - A(\mathrm{I} \odot A^2)$$
$$= \mathrm{J} \odot (A(\mathrm{J} \odot A^2)) + \underbrace{\mathrm{J} \odot (A(\mathrm{I} \odot A^2))}_{=A(\mathrm{I} \odot A^2)} - A(\mathrm{I} \odot A^2)$$
$$= \mathrm{J} \odot (A(\mathrm{J} \odot A^2)). \tag{13}$$

Secondly, we have that $A_{i,i}^2 = \sum_k A_{i,k}$. Thus lemma D.1 implies

$$(\mathrm{I} \odot A^2)A - A = A \odot (A\mathrm{J}). \tag{14}$$

From equality (13) and (14), we can conclude. $\square$

An alternative understanding of how $P_3^*$ computes the number of 3-paths connecting two nodes is illustrated in Figure 11. The process can be described as follows:

The expression $A(\mathrm{J} \odot A^2)$ calculates, from a given node, a non-closed 2-path followed by a 1-path. This computation inherently includes non-closed 3-paths as well as 3-cycles. The 3-cycles are subsequently removed by the Hadamard multiplication with J, which zeroes out the diagonal elements. However, this operation also allows the possibility of traversing a 2-path and then returning to the intermediate node. To account for this and eliminate such paths, we subtract the term $A \odot (A\mathrm{J})$.

Thanks to formula (12), we can derive the 4-cycle formula.

**Corollary D.2.1**
*The following formula, denoted as $C_{4f}$ computes the number of 4-cycles linking two nodes*

$$C_{4f} = A \odot (A(\mathrm{J} \odot A^2)) - A \odot (A\mathrm{J}). \tag{15}$$

In terms of time complexity, $P_3^*$ is more efficient than $P_3$. The ratio of time complexity of $P_3^*$ over $P_3$ is $\frac{1}{2}$. It is directly derived from the number of matrix multiplications in both formulas. Figure 7 shows the gain of complexity of $P_3^*$ and the ratio between the two formulas.

Surprisingly, even for $l = 3$, GRL allows to improve the computation of path and cycle at edge-level in terms of time complexity.

**4-paths and 5-cycles**   The most effective explicit formula discovered to date for calculating the number of 4-paths connecting two nodes was proposed by Voropaev and Perepechko (2012), it is

$$P_4 = \mathrm{J} \odot A^4 - \mathrm{J} \odot (A(\mathrm{I} \odot A^2)A) + 2(\mathrm{J} \odot A^2) \tag{16}$$
$$- (\mathrm{I} \odot A^2)(\mathrm{J} \odot A^2) - (\mathrm{J} \odot A^2)(\mathrm{I} \odot A^2)$$
$$- A(\mathrm{I} \odot A^3) - (\mathrm{I} \odot A^3)A + 3A \odot A^2.$$

Following the formula for the $l$-cycle proposed in Voropaev and Perepechko (2012), we obtain for the 5-cycle the following formula

$$C_5 = A \odot P_4 = A \odot A^4 - A \odot (A(I \odot A^2)A) \tag{17}$$
$$- (I \odot A^2)(A \odot A^2) - (A \odot A^2)(I \odot A^2)$$
$$- A(I \odot A^3) - (I \odot A^3)A + 5A \odot A^2.$$

Again, GRL found an improved formula for $l = 4$.

**Theorem D.3**
*The following formula, denoted as $P_4^*$, computes the number of 4-paths linking two nodes*

$$P_4^* = J \odot (A(J \odot (A(J \odot A^2)))) - J \odot (A(A \odot (AJ))) \tag{18}$$
$$- J \odot ((A \odot (AJ))A) - A \odot ((A \odot A^2)J) + 2A \odot A^2.$$

*Proof.* We will show that $P_4^* = P_4$. Firstly, we have

$$J \odot A^4 = J \odot ((A((J + I) \odot A^3)))$$
$$= J \odot (A(J \odot A^3)) + \underbrace{J \odot (A(I \odot A^3))}_{=A(I \odot A^3)}$$
$$= J \odot (A(J \odot (A((J + I) \odot A^2)))) + A(I \odot A^3)$$
$$= J \odot (A(J \odot (A(J \odot A^2)))) + \underbrace{J \odot (A(J \odot (A(I \odot A^2))))}_{=(J \odot A^2)(I \odot A^2)} + A(I \odot A^3)$$
$$= J \odot (A(J \odot (A(J \odot A^2)))) + (J \odot A^2)(I \odot A^2) + A(I \odot A^3). \tag{19}$$

Secondly, we have from equality (14)

$$J \odot ((A \odot (AJ))A) = J \odot ((I \odot A^2)A - A)A)$$
$$= \underbrace{J \odot ((I \odot A^2)A^2)}_{(I \odot A^2)(J \odot A^2)} - J \odot A^2$$
$$= (I \odot A^2)(J \odot A^2) - J \odot A^2. \tag{20}$$

Thirdly, we have from equality (14)

$$J \odot (A(A \odot (AJ))) = J \odot (A(I \odot A^2)A - A))$$
$$= J \odot (A(I \odot A^2)A) - J \odot A^2. \tag{21}$$

And eventually, we have $A_{i,i}^3 = \sum_k (A \odot A^2)_{i,k}$. Thus lemma D.1 implies

$$(I \odot A^3)A - A \odot A^2 = A \odot ((A \odot A^2)J). \tag{22}$$

From equality (19), (20),(21) and (22), we can conclude. □

An alternative understanding of how $P_4^*$ computes the number of 4-paths connecting two nodes is illustrated in Figure 12. The process can be described as follows:

The expression $A(J \odot (A(J \odot A^2))) - A(A \odot (AJ)) = AP_3^*$ calculates, from a given node, a non-closed 3-path followed by a 1-path. This computation inherently includes non-closed 4-paths as well as 4-cycles. The 4-cycles are subsequently removed by the Hadamard multiplication with J, which zeroes out the diagonal elements. However, this operation also allows the possibility of traversing a 3-path and then returning to an intermediate node. To account for this and eliminate respectively paths returning to the third and second nodes of the 3-paths, we subtract the terms $J \odot ((A \odot (AJ))A) - A \odot A^2$ and $A \odot ((A \odot A^2)J) - A \odot A^2$.

Thanks to formula (18), we can derive the 5-cycle formula.

**Corollary D.3.1**
*The following formula, denoted as $C_{5f}$ computes the number of 5-cycles linking two nodes*

$$C_{5f} = A \odot (A(J \odot (A(J \odot A^2)))) - A \odot (A(A \odot (AJ))) \tag{23}$$
$$- A \odot ((A \odot (AJ))A) - A \odot ((A \odot A^2)J) + 2A \odot A^2.$$

$$\text{J} \odot (AP_{3f}) \quad - \quad \text{J} \odot ((A \odot (A\text{J}))A) \quad - \quad A \odot ((A \odot A^2)\text{J}) \quad + \quad 2A \odot A^2 \quad = \quad P_{4f}$$

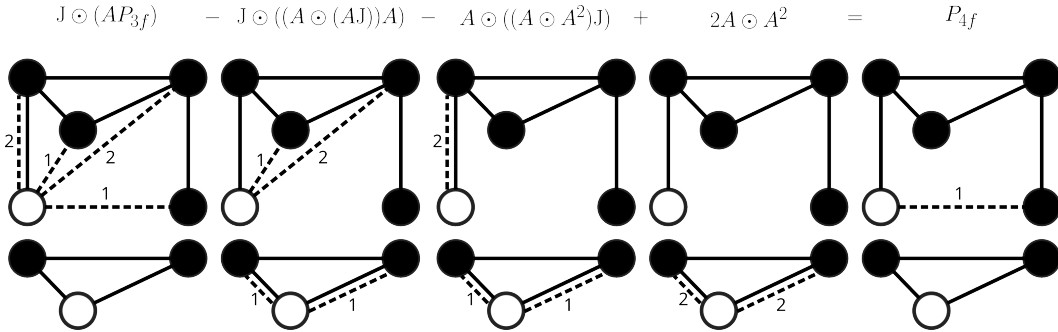

Figure 12: computation of $P_4^*$ for two graphs, the dashed lines indicate an entry in the column corresponding to the white node in the matrix associated with the term specified above.

In terms of time complexity, $P_4^*$ is more efficient than $P_4$. The ratio of time complexity of $P_4^*$ over $P_4$ is $\frac{4}{9}$. It is directly derived from the number of matrix multiplications in both formulas. Figure 7 shows the gain of complexity of $P_4^*$ and the ratio between the two formulas.

GRL improves the computation of path and cycle at edge-level for $l = 5$.

**5-paths and 6-cycles** The most effective explicit formula discovered to date for calculating the number of 5-paths connecting two nodes was proposed by Voropaev and Perepechko (2012), it is

$$
\begin{aligned}
P_5 = {}& \text{J} \odot A^5 - (\text{I} \odot A^4)A - A(\text{I} \odot A^4) - (\text{I} \odot A^3)(\text{J} \odot A^2) - (\text{J} \odot A^2)(\text{I} \odot A^3) \qquad (24)\\
& - (\text{I} \odot A^2)(\text{J} \odot A^3) - (\text{J} \odot A^3)(\text{I} \odot A^2) - \text{J} \odot (A(\text{I} \odot A^3)A) + 3A \odot A^3\\
& + 2A(\text{I} \odot A^2)(\text{I} \odot A^2) + 2(\text{I} \odot A^2)(\text{I} \odot A^2)A + 3A \odot A^2 \odot A^2 + (\text{I} \odot A^2)A(\text{I} \odot A^2)\\
& - \text{J} \odot (A(\text{I} \odot A^2)A^2) - \text{J} \odot (A^2(\text{I} \odot A^2)A) + 3\text{J} \odot ((A \odot A^2)A) + 3\text{J} \odot (A(A \odot A^2))\\
& + (\text{I} \odot (A(\text{I} \odot A^2)A))A + A(\text{I} \odot (A(\text{I} \odot A^2)A)) - 6(\text{I} \odot A^2)A - 6A(\text{I} \odot A^2)\\
& - 4A \odot A^2 + 3\text{J} \odot A^3 + 4A.
\end{aligned}
$$

Following the formula for the $l$-cycle proposed in Voropaev and Perepechko (2012), we obtain for the 6-cycle the following formula

$$
\begin{aligned}
C_6 = {}& A \odot A^5 - (\text{I} \odot A^4)A - A(\text{I} \odot A^4) - (\text{I} \odot A^3)(A \odot A^2) - (A \odot A^2)(\text{I} \odot A^3) \qquad (25)\\
& - (\text{I} \odot A^2)(A \odot A^3) - (A \odot A^3)(\text{I} \odot A^2) - A \odot (A(\text{I} \odot A^3)A) + 6A \odot A^3\\
& + 2A(\text{I} \odot A^2)(\text{I} \odot A^2) + 2(\text{I} \odot A^2)(\text{I} \odot A^2)A + 3A \odot A^2 \odot A^2 + (\text{I} \odot A^2)A(\text{I} \odot A^2)\\
& - A \odot (A(\text{I} \odot A^2)A^2) - A \odot (A^2(\text{I} \odot A^2)A) + 3A \odot ((A \odot A^2)A) + 3A \odot (A(A \odot A^2))\\
& + (\text{I} \odot (A(\text{I} \odot A^2)A))A + A(\text{I} \odot (A(\text{I} \odot A^2)A)) - 6(\text{I} \odot A^2)A - 6A(\text{I} \odot A^2)\\
& - 4A \odot A^2 + 4A.
\end{aligned}
$$

Again, GRL found an improved formula for $l = 5$.

**Theorem D.4**
*The following formula, denoted as $P_5^*$, computes the number of 5-paths linking two nodes*

$$
\begin{aligned}
P_5^* = {}& \text{J} \odot (AP_4^*) - (\text{J} \odot A^2) \odot ((A \odot A^2)\text{J}) - A \odot (C_{4f}\text{J}) - (A\text{J}) \odot P_3^* \qquad (26)\\
& + P_3^* + C_{4f} + 2A \odot A^2 \odot A^2 + 3\text{J} \odot ((A \odot A^2)A) - 4A \odot A^2.
\end{aligned}
$$

*Proof.* We will show that $P_5^* = P_5$. First, we have

$$J \odot A^5 = J \odot ((A((J + I) \odot A^4)))$$
$$= J \odot (A(J \odot A^4)) + \underbrace{J \odot (A(I \odot A^4))}_{=A(I \odot A^4)}$$
$$= J \odot (A(J \odot (A((J + I) \odot A^3)))) + A(I \odot A^4)$$
$$= J \odot (A(J \odot (A(J \odot A^3)))) + \underbrace{J \odot (A(J \odot (A(I \odot A^3))))}_{=(J \odot A^2)(I \odot A^3)} + A(I \odot A^4)$$
$$= J \odot (A(J \odot (A(J \odot (A((J + I) \odot A^2)))))) + (J \odot A^2)(I \odot A^3) + A(I \odot A^4)$$
$$= J \odot (A(J \odot (A(J \odot (A(J \odot A^2)))))) + J \odot (A(J \odot (A(J \odot (A(I \odot A^2)))))) \quad (27)$$
$$+ (J \odot A^2)(I \odot A^3) + A(I \odot A^4).$$

And

$$J \odot (A(J \odot (A(J \odot (A(I \odot A^2)))))) = J \odot (A(J \odot A^2)(I \odot A^2))$$
$$= \underbrace{J \odot (A^3(I \odot A^2))}_{=(J \odot A^3)(I \odot A^2)} - \underbrace{J \odot (A(I \odot A^2)(I \odot A^2))}_{=A(I \odot A^2)(I \odot A^2)}$$
$$= (J \odot A^3)(I \odot A^2) - A(I \odot A^2)(I \odot A^2). \quad (28)$$

Thus equation (27) and (28) give

$$J \odot A^5 = J \odot (A(J \odot (A(J \odot (A(J \odot A^2)))))) + (J \odot A^3)(I \odot A^2) + (J \odot A^2)(I \odot A^3) \quad (29)$$
$$+ A(I \odot A^4) - A(I \odot A^2)(I \odot A^2).$$

Second, we have from equality (20)

$$J \odot (A(J \odot ((A \odot (AJ))A))) = J \odot (A((I \odot A^2)(J \odot A^2) - J \odot A^2))$$
$$= J \odot (A(I \odot A^2)A^2) - \underbrace{J \odot (A(I \odot A^2)(I \odot A^2))}_{=A(I \odot A^2)(I \odot A^2)} - \underbrace{J \odot (A(J \odot A^2))}_{=J \odot A^3 - A(I \odot A^2)}$$
$$= J \odot (A(I \odot A^2)A^2) - A(I \odot A^2)(I \odot A^2) - J \odot A^3 \quad (30)$$
$$+ A(I \odot A^2).$$

Third, we have from equality (21)

$$J \odot (A(J \odot (A(A \odot (AJ))))) = J \odot (A(J \odot (A(I \odot A^2)A) - J \odot A^2))$$
$$= J \odot (A^2(I \odot A^2)A) - \underbrace{J \odot (A(I \odot (A(I \odot A^2)A))}_{=A(I \odot (A(I \odot A^2)A))} - \underbrace{J \odot (A(J \odot A^2))}_{=J \odot A^3 - A(I \odot A^2)}$$
$$= J \odot (A^2(I \odot A^2)A) - A(I \odot (A(I \odot A^2)A) - J \odot A^3 \quad (31)$$
$$+ A(I \odot A^2).$$

Fourth, we have from equality (22)

$$J \odot (A(A \odot ((A \odot A^2)J))) = J \odot (A((I \odot A^3)A) - A \odot A^2))$$
$$= J \odot (A(I \odot A^3)A) - J \odot (A(A \odot A^2)). \quad (32)$$

Fifth, we have $A_{i,i}^3 = \sum_k (A \odot A^2)_{i,k}$. Thus lemma D.1 implies

$$(I \odot A^3)(J \odot A^2) - A \odot A^2 \odot A^2 = (J \odot A^2) \odot ((A \odot A^2)J). \quad (33)$$

Sixth, we have $(AP_3^*)_{i,i} = \sum_k (C_{4f})_{i,k}$. Thus lemma D.1 implies

$$(I \odot (AP_3^*))A - C_{4f} = A \odot (C_{4f}J). \quad (34)$$

Eventually, we have $(A^2)_{i,i} = \sum_k A_{i,k}$. Thus lemma D.1 implies

$$(I \odot A^2)P_3^* - A \odot P_3^* = P_3^* \odot (AJ). \quad (35)$$

By removing equality (29), (30), (31), (32), (33),(34), (35) and $2J \odot (A(A \odot A^2))$ to $P_5$, we obtain exactly $P_3^* + C_{4f} + 2A \odot A^2 \odot A^2 + 3J \odot ((A \odot A^2)A) - 4A \odot A^2$. It concludes the proof. $\square$

An alternative understanding of how $P_5^*$ computes the number of 5-paths connecting two nodes is illustrated in Figure 13. The process can be described as follows:

The expression $AP_4^*$ calculates, from a given node, a non-closed 4-path followed by a 1-path. This computation inherently includes non-closed 5-paths as well as 5-cycles. The 5-cycles are subsequently removed by the Hadamard multiplication with J, which zeroes out the diagonal elements. However, this operation also allows the possibility of traversing a 4-path and then returning to an intermediate node. To account for this and eliminate respectively paths returning to the fourth, third and second nodes of the 4-paths, we subtract the terms $A \odot ((C_{4f})\mathrm{J}) - C_{4f}$, $(\mathrm{J} \odot A^2) \odot ((A \odot A^2)\mathrm{J}) + 4A \odot A^2 - 2A \odot A^2 \odot A^2 - 3\mathrm{J} \odot ((A \odot A^2)A)$ and $(A\mathrm{J}) \odot P_3^* - P_3^*$.

Thanks to formula (26), we can derive the 6-cycle formula.

**Corollary D.4.1**
*The following formula, denoted as $C_{6f}$ computes the number of 6-cycles linking two nodes*

$$C_6^* = A \odot (AP_4^*) - (A \odot A^2) \odot ((A \odot A^2)\mathrm{J}) - A \odot ((C_{4f})\mathrm{J}) - (A\mathrm{J}) \odot C_{4f} \qquad (36)$$
$$+ 2C_{4f} + 2A \odot A^2 \odot A^2 + 3A \odot ((A \odot A^2)A) - 4A \odot A^2.$$

In terms of time complexity, $P_5^*$ is more efficient than $P_5$. The ratio of time complexity of $P_5^*$ over $P_5$ is $\frac{1}{4}$. It is directly derived from the number of matrix multiplications in both formulas. The significant decrease in time complexity can be attributed to the presence of both $P_4^*$, $C_{4f}$ and $P_3^*$ in the computational formula. The contributions of these terms to the time complexity are cumulative, meaning that each occurrence of $P_4^*$, $C_{4f}$ and $P_3^*$ adds to the total computational gain. As a result, their individual time complexities are aggregated, leading to the observed diminution in the overall time complexity of the formula. Figure 7 shows the gain of complexity of $P_5^*$ and the ratio between the two formulas.

GRL improves the computation of path and cycle at edge-level for $l = 5$ by a factor 4.

**6-paths and 7-cycles** The most effective explicit formula discovered to date for calculating the number of 6-paths connecting two nodes was proposed by Voropaev and Perepechko (2012), it is

$$P_6 = \mathrm{J} \odot A^6 - (\mathrm{I} \odot A^5)A - A(\mathrm{I} \odot A^5) - (\mathrm{I} \odot A^2)(\mathrm{J} \odot A^4) - (\mathrm{J} \odot A^4)(\mathrm{I} \odot A^2) \qquad (37)$$
$$- (\mathrm{I} \odot A^4)(\mathrm{J} \odot A^2) - (\mathrm{J} \odot A^2)(\mathrm{I} \odot A^4) - \mathrm{J} \odot (A(\mathrm{I} \odot A^4)A) + 3A \odot A^4$$
$$- (\mathrm{J} \odot A^3)(\mathrm{I} \odot A^3) - (\mathrm{I} \odot A^3)(\mathrm{J} \odot A^3) - \mathrm{J} \odot (A(\mathrm{I} \odot A^2)A^3) - \mathrm{J} \odot (A^3(\mathrm{I} \odot A^2)A)$$
$$- \mathrm{J} \odot (A(\mathrm{I} \odot A^3)A^2) - \mathrm{J} \odot (A^2(\mathrm{I} \odot A^3)A) + 4A(\mathrm{I} \odot A^2)(\mathrm{I} \odot A^3) + 4(\mathrm{I} \odot A^3)(\mathrm{I} \odot A^2)A$$
$$+ 6A \odot A^2 \odot A^3 + (\mathrm{I} \odot A^2)A(\mathrm{I} \odot A^3) + (\mathrm{I} \odot A^3)A(\mathrm{I} \odot A^2) + 3\mathrm{J} \odot ((A \odot A^3)A)$$
$$+ 3\mathrm{J} \odot (A(A \odot A^3)) + (\mathrm{I} \odot (A(\mathrm{I} \odot A^3)A))A + A(\mathrm{I} \odot (A(\mathrm{I} \odot A^3)A)) - \mathrm{J} \odot (A^2(\mathrm{I} \odot A^2)A^2)$$
$$+ 2A(\mathrm{I} \odot A^2)(\mathrm{I} \odot A^2) + 2(\mathrm{I} \odot A^2)(\mathrm{I} \odot A^2)A + \mathrm{J} \odot A^2 \odot A^2 \odot A^2 + (\mathrm{I} \odot A^2)(\mathrm{J} \odot A^2)(\mathrm{I} \odot A^2)$$
$$+ 3\mathrm{J} \odot ((A \odot A^2)A^2) + 3\mathrm{J} \odot (A^2(A \odot A^2)) + (\mathrm{I} \odot (A(\mathrm{I} \odot A^2)A))(\mathrm{J} \odot A^2) + (\mathrm{J} \odot A^2)(\mathrm{I} \odot (A(\mathrm{I} \odot A^2)A))$$
$$+ \mathrm{J} \odot ((\mathrm{I} \odot A^2)A(\mathrm{I} \odot A^2)A) + \mathrm{J} \odot (A(\mathrm{I} \odot A^2)A(\mathrm{I} \odot A^2)) + 2\mathrm{J} \odot (A(\mathrm{I} \odot A^2)(\mathrm{I} \odot A^2)A)$$
$$+ (\mathrm{I} \odot (A(\mathrm{I} \odot A^2)A^2))A + A(\mathrm{I} \odot (A(\mathrm{I} \odot A^2)A^2)) + (\mathrm{I} \odot (A^2(\mathrm{I} \odot A^2)A))A + A(\mathrm{I} \odot (A^2(\mathrm{I} \odot A^2)A))$$
$$+ 3\mathrm{J} \odot ((A \odot A^2 \odot A^2)A) + 3\mathrm{J} \odot (A(A \odot A^2 \odot A^2)) - 12(\mathrm{I} \odot A^2)(A \odot A^2) - 12(A \odot A^2)(\mathrm{I} \odot A^2)$$
$$- 4\mathrm{J} \odot A^2 \odot A^2 - 8A \odot (A(A \odot A^2)) - 8A \odot ((A \odot A^2)A) - 3A \odot (A(\mathrm{I} \odot A^2)A)$$
$$+ 3\mathrm{J} \odot (A(A \odot A^2)A) + \mathrm{J} \odot (A(\mathrm{I} \odot (A(\mathrm{I} \odot A^2)A))A) - 4\mathrm{J} \odot (A(A \odot A^2)) - 4\mathrm{J} \odot ((A \odot A^2)A)$$
$$+ 4\mathrm{J} \odot A^4 - 5A(\mathrm{I} \odot A^3) - 5(\mathrm{I} \odot A^3)A - 4(\mathrm{I} \odot (A(A \odot A^2)))A - 4A(\mathrm{I} \odot (A(A \odot A^2)))$$
$$- 4(\mathrm{I} \odot ((A \odot A^2)A))A - 4A(\mathrm{I} \odot ((A \odot A^2)A)) - 7(\mathrm{I}A^2)(\mathrm{J} \odot A^2) - 7(\mathrm{J} \odot A^2)(\mathrm{I}A^2)$$
$$- 10\mathrm{J} \odot (A(\mathrm{I} \odot A^2)A) + 44A \odot A^2 + 12\mathrm{J} \odot A^2.$$

$$\text{J} \odot (AP_{4f}) - (\text{J} \odot A^2) \odot ((A \odot A^2)\text{J}) - A \odot ((C_{4f})\text{J}) \quad - \quad (A\text{J}) \odot P_{3f} \qquad + \qquad P_{3f}$$

$$+ \quad C_{4f} \quad - \quad 4A \odot A^2 \quad + \quad 2A \odot A^2 \odot A^2 \quad + 3\text{J} \odot ((A \odot A^2)A) = \qquad P_{5f}$$

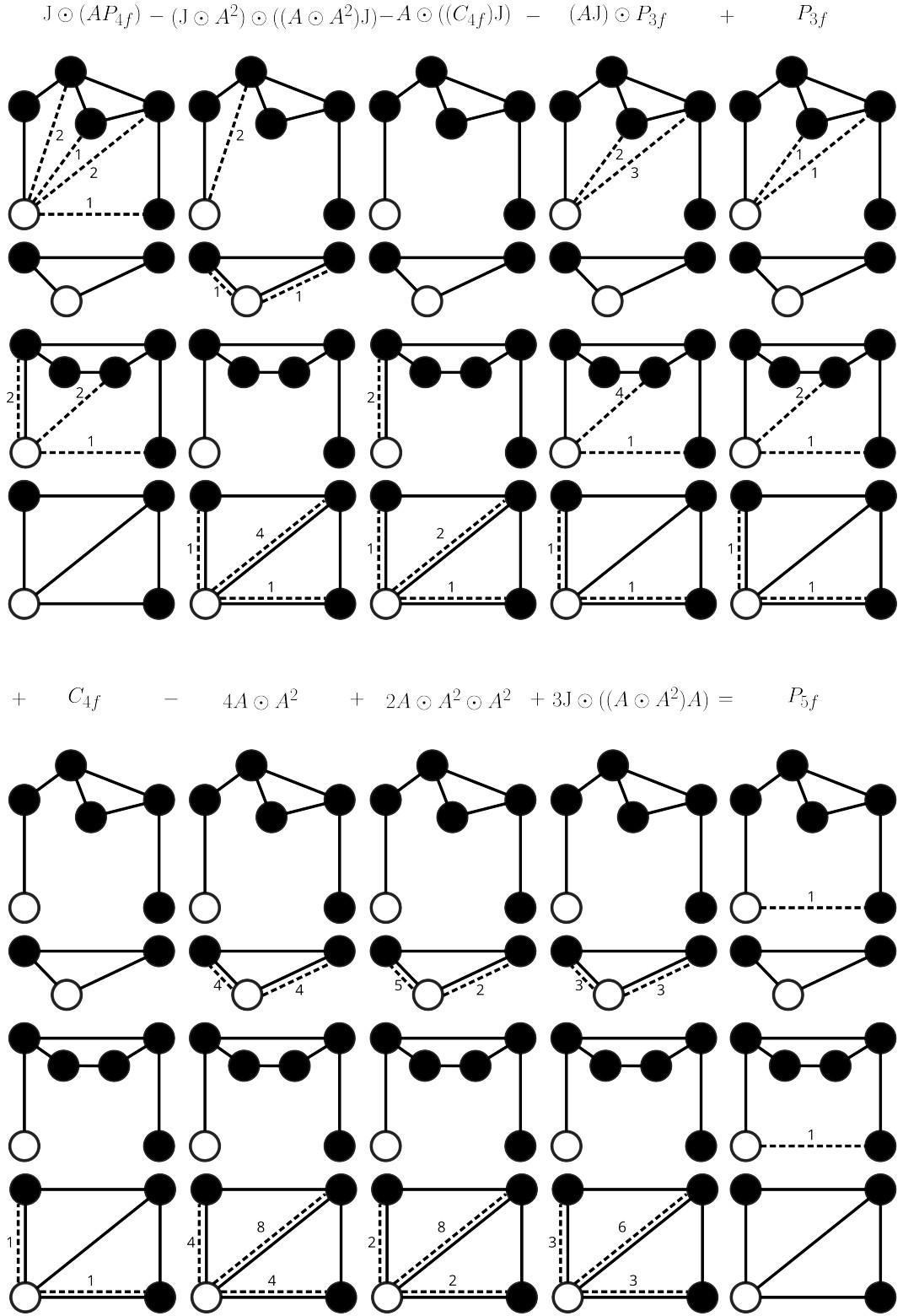

Figure 13: computation of $P_5^*$ for four graphs, the dashed lines indicate an entry in the column corresponding to the white node in the matrix associated with the term specified above.

Following the formula for the $l$-cycle proposed in Voropaev and Perepechko (2012), we obtain for the 7-cycle the following formula

$$
\begin{aligned}
C_7 = {}& A \odot A^6 - (\mathrm{I} \odot A^5)A - A(\mathrm{I} \odot A^5) - (\mathrm{I} \odot A^2)(A \odot A^4) - (A \odot A^4)(\mathrm{I} \odot A^2) \qquad (38) \\
& - (\mathrm{I} \odot A^4)(A \odot A^2) - (A \odot A^2)(\mathrm{I} \odot A^4) - A \odot (A(\mathrm{I} \odot A^4)A) + 3A \odot A^4 \\
& - (A \odot A^3)(\mathrm{I} \odot A^3) - (\mathrm{I} \odot A^3)(A \odot A^3) - A \odot (A(\mathrm{I} \odot A^2)A^3) - A \odot (A^3(\mathrm{I} \odot A^2)A) \\
& - A \odot (A(\mathrm{I} \odot A^3)A^2) - A \odot (A^2(\mathrm{I} \odot A^3)A) + 4A(\mathrm{I} \odot A^2)(\mathrm{I} \odot A^3) + 4(\mathrm{I} \odot A^3)(\mathrm{I} \odot A^2)A \\
& + 6A \odot A^2 \odot A^3 + (\mathrm{I} \odot A^2)A(\mathrm{I} \odot A^3) + (\mathrm{I} \odot A^3)A(\mathrm{I} \odot A^2) + 3A \odot ((A \odot A^3)A) \\
& + 3A \odot (A(A \odot A^3)) + (\mathrm{I} \odot (A(\mathrm{I} \odot A^3)A))A + A(\mathrm{I} \odot (A(\mathrm{I} \odot A^3)A)) - A \odot (A^2(\mathrm{I} \odot A^2)A^2) \\
& + 2A(\mathrm{I} \odot A^2)(\mathrm{I} \odot A^2) + 2(\mathrm{I} \odot A^2)(\mathrm{I} \odot A^2)A + A \odot A^2 \odot A^2 \odot A^2 + (\mathrm{I} \odot A^2)(A \odot A^2)(\mathrm{I} \odot A^2) \\
& + 3A \odot ((A \odot A^2)A^2) + 3A \odot (A^2(A \odot A^2)) + (\mathrm{I} \odot (A(\mathrm{I} \odot A^2)A))(A \odot A^2) + (A \odot A^2)(\mathrm{I} \odot (A(\mathrm{I} \odot A^2)A)) \\
& + A \odot ((\mathrm{I} \odot A^2)A(\mathrm{I} \odot A^2)A) + A \odot (A(\mathrm{I} \odot A^2)A(\mathrm{I} \odot A^2)) + 2A \odot (A(\mathrm{I} \odot A^2)(\mathrm{I} \odot A^2)A) \\
& + (\mathrm{I} \odot (A(\mathrm{I} \odot A^2)A^2))A + A(\mathrm{I} \odot (A(\mathrm{I} \odot A^2)A^2)) + (\mathrm{I} \odot (A^2(\mathrm{I} \odot A^2)A))A + A(\mathrm{I} \odot (A^2(\mathrm{I} \odot A^2)A)) \\
& + 3A \odot ((A \odot A^2 \odot A^2)A) + 3A \odot (A(A \odot A^2 \odot A^2)) - 12(\mathrm{I} \odot A^2)(A \odot A^2) - 12(A \odot A^2)(\mathrm{I} \odot A^2) \\
& - 4A \odot A^2 \odot A^2 - 8A \odot (A(A \odot A^2)) - 8A \odot ((A \odot A^2)A) - 3A \odot (A(\mathrm{I} \odot A^2)A) \\
& + 3A \odot (A(A \odot A^2)A) + A \odot (A(\mathrm{I} \odot (A(\mathrm{I} \odot A^2)A))A) - 4A \odot (A(A \odot A^2)) - 4A \odot ((A \odot A^2)A) \\
& + 4A \odot A^4 - 5A(\mathrm{I} \odot A^3) - 5(\mathrm{I} \odot A^3)A - 4(\mathrm{I} \odot (A(A \odot A^2)))A - 4A(\mathrm{I} \odot (A(A \odot A^2))) \\
& - 4(\mathrm{I} \odot ((A \odot A^2)A))A - 4A(\mathrm{I} \odot ((A \odot A^2)A)) - 7(\mathrm{I}A^2)(A \odot A^2) - 7(A \odot A^2)(\mathrm{I}A^2) \\
& - 10A \odot (A(\mathrm{I} \odot A^2)A) + 56A \odot A^2.
\end{aligned}
$$

Again, GRL found an improved formula for $l = 6$.

**Theorem D.5**
*The following formula, denoted as $P_6^*$ computes the number of 6-paths linking two nodes*

$$
\begin{aligned}
P_6^* = {}& \mathrm{J} \odot (AP_5^*) - P_3^* \odot ((A \odot A^2)\mathrm{J}) - A \odot (C_{5f}\mathrm{J}) - P_4^* \odot (A\mathrm{J}) \qquad (39) \\
& + P_4^* + C_{5f} - (\mathrm{J} \odot A^2) \odot (C_{4f}\mathrm{J}) + 4A \odot A^2 \odot P_3^* + 3\mathrm{J} \odot ((A \odot A^2)(\mathrm{J} \odot A^2)) \\
& + \mathrm{J} \odot A^2 \odot A^2 \odot A^2 + 3\mathrm{J} \odot ((A \odot A^2 \odot A^2)A) - 4\mathrm{J} \odot A^2 \odot A^2 - 8A \odot (A(A \odot A^2)) \\
& - 8A \odot ((A \odot A^2)A) - 4\mathrm{J} \odot ((A \odot A^2)A) - 3A \odot ((A \odot A^2)\mathrm{J}) + 17A \odot A^2 + 3\mathrm{J} \odot A^2.
\end{aligned}
$$

*Proof.* We will show that $P_6^* = P_6$. First, we have

$$
\begin{aligned}
\mathrm{J} \odot A^6 &= \mathrm{J} \odot ((A((\mathrm{J} + \mathrm{I}) \odot A^5))) \\
&= \mathrm{J} \odot (A(\mathrm{J} \odot A^5)) + \underbrace{\mathrm{J} \odot (A(\mathrm{I} \odot A^5))}_{=A(\mathrm{I} \odot A^5)} \\
&= \mathrm{J} \odot (A(\mathrm{J} \odot (A((\mathrm{J} + \mathrm{I}) \odot A^4)))) + A(\mathrm{I} \odot A^5) \\
&= \mathrm{J} \odot (A(\mathrm{J} \odot (A(\mathrm{J} \odot A^4)))) + \underbrace{\mathrm{J} \odot (A(\mathrm{J} \odot (A(\mathrm{I} \odot A^4))))}_{=(\mathrm{J} \odot A^2)(\mathrm{I} \odot A^4)} + A(\mathrm{I} \odot A^5) \\
&= \mathrm{J} \odot (A(\mathrm{J} \odot (A(\mathrm{J} \odot (A((\mathrm{J} + \mathrm{I}) \odot A^3)))))) + (\mathrm{J} \odot A^2)(\mathrm{I} \odot A^4) + A(\mathrm{I} \odot A^5) \\
&= \mathrm{J} \odot (A(\mathrm{J} \odot (A(\mathrm{J} \odot (A(\mathrm{J} \odot A^3)))))) + \underbrace{\mathrm{J} \odot (A(\mathrm{J} \odot (A(\mathrm{J} \odot (A(\mathrm{I} \odot A^3))))))}_{=(\mathrm{J} \odot A^3)(\mathrm{I} \odot A^3) - A(\mathrm{I} \odot A^2)(\mathrm{I} \odot A^3)} \\
&\quad + (\mathrm{J} \odot A^2)(\mathrm{I} \odot A^4) + A(\mathrm{I} \odot A^5) \\
&= \mathrm{J} \odot (A(\mathrm{J} \odot (A(\mathrm{J} \odot (A(\mathrm{J} \odot (A((\mathrm{J} + \mathrm{I}) \odot A^2)))))))) + (\mathrm{J} \odot A^3)(\mathrm{I} \odot A^3) \\
&\quad - A(\mathrm{I} \odot A^2)(\mathrm{I} \odot A^3) + (\mathrm{J} \odot A^2)(\mathrm{I} \odot A^4) + A(\mathrm{I} \odot A^5) \\
&= \mathrm{J} \odot (A(\mathrm{J} \odot (A(\mathrm{J} \odot (A(\mathrm{J} \odot (A(\mathrm{J} \odot A^2)))))))) + \underbrace{\mathrm{J} \odot (A(\mathrm{J} \odot (A(\mathrm{J} \odot (A(\mathrm{J} \odot (A(\mathrm{I} \odot A^2))))))))}_{=(\mathrm{J} \odot A^4)(\mathrm{I} \odot A^2) - A(\mathrm{I} \odot A^2)(\mathrm{I} \odot A^3) - (\mathrm{J} \odot A^2)(\mathrm{I} \odot A^2)(\mathrm{I} \odot A^2)} \\
&\quad + (\mathrm{J} \odot A^3)(\mathrm{I} \odot A^3) - A(\mathrm{I} \odot A^2)(\mathrm{I} \odot A^3) + (\mathrm{J} \odot A^2)(\mathrm{I} \odot A^4) + A(\mathrm{I} \odot A^5) \\
&= \mathrm{J} \odot (A(\mathrm{J} \odot (A(\mathrm{J} \odot (A(\mathrm{J} \odot (A(\mathrm{J} \odot A^2)))))))) + (\mathrm{J} \odot A^4)(\mathrm{I} \odot A^2) \qquad (40) \\
&\quad - 2A(\mathrm{I} \odot A^2)(\mathrm{I} \odot A^3) - (\mathrm{J} \odot A^2)(\mathrm{I} \odot A^2)(\mathrm{I} \odot A^2) + (\mathrm{J} \odot A^3)(\mathrm{I} \odot A^3) \\
&\quad + (\mathrm{J} \odot A^2)(\mathrm{I} \odot A^4) + A(\mathrm{I} \odot A^5).
\end{aligned}
$$

Second, we have from equality (30)

$$
\begin{aligned}
\mathrm{J} \odot (A(\mathrm{J} \odot (A(\mathrm{J} \odot ((A \odot (A\mathrm{J}))A))))) &= \mathrm{J} \odot (A((\mathrm{J} \odot (A(\mathrm{I} \odot A^2)A^2) - A(\mathrm{I} \odot A^2)(\mathrm{I} \odot A^2) \\
&\qquad - \mathrm{J} \odot A^3 + A(\mathrm{I} \odot A^2))) \\
&= \underbrace{\mathrm{J} \odot (A(\mathrm{J} \odot (A(\mathrm{I} \odot A^2)A^2)))}_{=\mathrm{J} \odot (A^2(\mathrm{I} \odot A^2)A^2) - A(\mathrm{I} \odot (A(\mathrm{I} \odot A^2)A^2))} - (\mathrm{J} \odot A^2)(\mathrm{I} \odot A^2)(\mathrm{I} \odot A^2) \\
&\quad \underbrace{\mathrm{J} \odot (A(\mathrm{J} \odot A^3))}_{=\mathrm{J} \odot A^4 - A(\mathrm{I} \odot A^3)} + (\mathrm{J} \odot A^2)(\mathrm{I} \odot A^2) \\
&= \mathrm{J} \odot (A^2(\mathrm{I} \odot A^2)A^2) - A(\mathrm{I} \odot (A(\mathrm{I} \odot A^2)A^2)) \quad (41) \\
&\quad - (\mathrm{J} \odot A^2)(\mathrm{I} \odot A^2)(\mathrm{I} \odot A^2) - \mathrm{J} \odot A^4 + A(\mathrm{I} \odot A^3) \\
&\quad + (\mathrm{J} \odot A^2)(\mathrm{I} \odot A^2).
\end{aligned}
$$

Third, we have from equalities (31)

$$
\begin{aligned}
\mathrm{J} \odot (A(\mathrm{J} \odot (A(\mathrm{J} \odot (A(A \odot (A\mathrm{J})))))))) &= \mathrm{J} \odot (A(\mathrm{J} \odot (A^2(\mathrm{I} \odot A^2)A) - A(\mathrm{I} \odot (A(\mathrm{I} \odot A^2)A)) \\
&\qquad - \mathrm{J} \odot A^3 + A(\mathrm{I} \odot A^2))) \\
&= \underbrace{\mathrm{J} \odot (A(\mathrm{J} \odot (A^2(\mathrm{I} \odot A^2)A)))}_{=\mathrm{J} \odot (A^3(\mathrm{I} \odot A^2)A) - A(\mathrm{I}(A^2(\mathrm{I} \odot A^2)A))} - (\mathrm{J} \odot A^2)(\mathrm{I} \odot (A(\mathrm{I} \odot A^2)A)) \\
&\quad - \mathrm{J} \odot A^4 + A(\mathrm{I} \odot A^3) + (\mathrm{J} \odot A^2)(\mathrm{I} \odot A^2) \\
&= \mathrm{J} \odot (A^3(\mathrm{I} \odot A^2)A) - A(\mathrm{I}(A^2(\mathrm{I} \odot A^2)A)) \qquad (42) \\
&\quad - (\mathrm{J} \odot A^2)(\mathrm{I} \odot (A(\mathrm{I} \odot A^2)A)) - \mathrm{J} \odot A^4 + A(\mathrm{I} \odot A^3) \\
&\quad + (\mathrm{J} \odot A^2)(\mathrm{I} \odot A^2).
\end{aligned}
$$

Fourth, we have from equality (32)

$$J \odot (A(J \odot (A(A \odot ((A \odot A^2)J))))) = J \odot (A(J \odot (A(I \odot A^3)A) - J \odot (A(A \odot A^2))))$$
$$= \underbrace{J \odot (A(J \odot (A(I \odot A^3)A)))}_{=J \odot (A^2(I \odot A^2)A) - A(I \odot (A(I \odot A^3)A))} - \underbrace{J \odot (A(J \odot (A(A \odot A^2))))}_{=J \odot (A^2(A \odot A^2)) - A(I \odot (A(A \odot A^2)))}$$
$$= J \odot (A^2(I \odot A^2)A) - A(I \odot (A(I \odot A^3)A)) \qquad (43)$$
$$- J \odot (A^2(A \odot A^2)) + A(I \odot (A(A \odot A^2))).$$

Fifth, we have from equality (33)

$$J \odot (A((J \odot A^2) \odot ((A \odot A^2)J))) = J \odot (A((I \odot A^3)(J \odot A^2) - A \odot A^2 \odot A^2))$$
$$= \underbrace{J \odot (A(I \odot A^3)(J \odot A^2))}_{J \odot (A(I \odot A^3)A^2) - A(I \odot A^3)(I \odot A^2)} - J \odot A(A \odot A^2 \odot A^2)$$
$$= J \odot (A(I \odot A^3)A^2) - A(I \odot A^3)(I \odot A^2) \qquad (44)$$
$$- J \odot A(A \odot A^2 \odot A^2).$$

Sixth, we have from equality (34)

$$J \odot (A(A \odot (C_{4f}J)) = J \odot (A((I \odot (AP_3))A - C_4))$$
$$= J \odot (A(I \odot (A(J \odot A^3 - A(I \odot A^2) - (I \odot A^2)A + A)))A)$$
$$- J \odot (A(A \odot A^3 - A(I \odot A^2) - (I \odot A^2)A + A))$$
$$= J \odot (A(I \odot A^4)A) - J \odot (A(I \odot A^2)(I \odot A^2)A) \qquad (45)$$
$$- J \odot (A(I \odot (A(I \odot A^2)A))A) + 2J \odot (A(I \odot A^2)A)$$
$$- J \odot (A(A \odot A^3)) + (J \odot A^2)(I \odot A^2) - (J \odot A^2).$$

Seventh, we have from equality (35)

$$J \odot (A(P_3^* \odot (AJ))) = J \odot (A((I \odot A^2)P_3 - C_4))$$
$$= J \odot (A((I \odot A^2)(J \odot A^3 - A(I \odot A^2) - (I \odot A^2)A + A)))$$
$$- J \odot (A(A \odot A^3 - A(I \odot A^2) - (I \odot A^2)A + A))$$
$$= J \odot (A(I \odot A^2)A^3) - A(I \odot A^2)(I \odot A^3) \qquad (46)$$
$$- J \odot (A(I \odot A^2)A(I \odot A^2)) - J \odot (A(I \odot A^2)(I \odot A^2)A)$$
$$+ 2J \odot (A(I \odot A^2)A) - J \odot (A(A \odot A^3)) + (J \odot A^2)(I \odot A^2) - (J \odot A^2).$$

Eighth, we have

$$J \odot (AP_3^*) = J \odot (AP_3)$$
$$= J \odot (A(J \odot A^3 - A(I \odot A^2) - (I \odot A^2)A + A)))$$
$$= J \odot A^4 - A(I \odot A^3) - (J \odot A^2)(I \odot A^2) - J \odot (A(I \odot A^2)A) + (J \odot A^2). \quad (47)$$

Ninth, we have

$$J \odot (AC_{4f}) = J \odot (AC_4)$$
$$= J \odot (A(A \odot A^3 - A(I \odot A^2) - (I \odot A^2)A + A)))$$
$$= J \odot (A(A \odot A^3)) - (J \odot A^2)(I \odot A^2) - J \odot (A(I \odot A^2)A) + (J \odot A^2). \quad (48)$$

Tenth, we have

$$J \odot (A(J \odot ((A \odot A^2)A))) = J \odot (A(A \odot A^2)A) - A(I \odot ((A \odot A^2)A)). \qquad (49)$$

Eleventh, we have

$$J \odot (A(J \odot (A(A \odot A^2)))) = J \odot (A^2(A \odot A^2)) - A(I \odot (A(A \odot A^2))). \qquad (50)$$

From equalities (40) to (50) combined with $J \odot (A(A \odot A^2 \odot A^2))$ and $J \odot (A(A \odot A^2))$ we obtain the equivalence between $J \odot (AP_5^*)$ and all those matrices.

Twelfth, we have $A^3_{i,i} = \sum_k (A \odot A^2)_{i,k}$. Thus lemma D.1 implies

$$P_3^* \odot ((A \odot A^2)\mathrm{J}) = (\mathrm{I} \odot A^3)P_3 - A^2 \odot C_4$$
$$= (\mathrm{I} \odot A^3)(\mathrm{J} \odot A^3) - (\mathrm{I} \odot A^3)A(\mathrm{I} \odot A^2) - (\mathrm{I} \odot A^3)(\mathrm{I} \odot A^2)A + (\mathrm{I} \odot A^3)A$$
$$- A^2 \odot A \odot A^3 + \underbrace{A^2 \odot (A(\mathrm{I} \odot A^2))}_{=(A \odot A^2)(\mathrm{I} \odot A^2)} + \underbrace{A^2 \odot ((\mathrm{I} \odot A^2)A)}_{=(\mathrm{I} \odot A^2)(A \odot A^2)} - A^2 \odot A$$
$$= (\mathrm{I} \odot A^3)(\mathrm{J} \odot A^3) - (\mathrm{I} \odot A^3)A(\mathrm{I} \odot A^2) - (\mathrm{I} \odot A^3)(\mathrm{I} \odot A^2)A \tag{51}$$
$$+ (\mathrm{I} \odot A^3)A - A \odot A^2 \odot A^3 + (A \odot A^2)(\mathrm{I} \odot A^2) + (\mathrm{I} \odot A^2)(A \odot A^2)$$
$$- A \odot A^2.$$

Thirteenth, we have $(AP_4^*)_{i,i} = \sum_k (C_{5f})_{i,k}$. Thus lemma D.1 implies

$$A \odot (C_{5f}\mathrm{J}) = (\mathrm{I} \odot (AP_4))A - C_5$$
$$= \underbrace{(\mathrm{I} \odot (A(\mathrm{J} \odot A^4)))A}_{=(\mathrm{I} \odot A^5)A} - \underbrace{(\mathrm{I} \odot (A(\mathrm{J} \odot (A(\mathrm{I} \odot A^2)A))))A}_{=(\mathrm{I} \odot (A^2(\mathrm{I} \odot A^2)A))A} + 2\underbrace{(\mathrm{I} \odot (A(\mathrm{J} \odot A^2)))A}_{=(\mathrm{I} \odot A^3)A}$$
$$- \underbrace{(\mathrm{I} \odot (A(\mathrm{I} \odot A^2)(\mathrm{J} \odot A^2)))A}_{=(\mathrm{I} \odot (A(\mathrm{I} \odot A^2)A^2))A} - \underbrace{(\mathrm{I} \odot (A(\mathrm{J} \odot A^2)(\mathrm{I} \odot A^2)))A}_{=(\mathrm{I} \odot A^3)(\mathrm{I} \odot A^2)A}$$
$$- (\mathrm{I} \odot (A(\mathrm{I} \odot A^3)A))A - (\mathrm{I} \odot A^2)(\mathrm{I} \odot A^3)A + 3(\mathrm{I} \odot (A(A \odot A^2)))A - A \odot A^4$$
$$+ A \odot (A(\mathrm{I} \odot A^2)A) + (\mathrm{I} \odot A^2)(A \odot A^2) + (A \odot A^2)(\mathrm{I} \odot A^2)$$
$$+ A(\mathrm{I} \odot A^3) + (\mathrm{I} \odot A^3)A - 5A \odot A^2$$

$$\tag{52}$$

$$- 2(\mathrm{I} \odot A^3)(\mathrm{I} \odot A^2)A - (\mathrm{I} \odot (A(\mathrm{I} \odot A^3)A))A + 3(\mathrm{I} \odot (A(A \odot A^2)))A$$
$$+ A \odot (A(\mathrm{I} \odot A^2)A) + (\mathrm{I} \odot A^2)(A \odot A^2) + (A \odot A^2)(\mathrm{I} \odot A^2) + 2(\mathrm{I} \odot A^3)A$$
$$+ A(\mathrm{I} \odot A^3) + (\mathrm{I} \odot A^3)A - 5A \odot A^2.$$

Fourteenth, we have $(A^2)_{i,i} = \sum_k A_{i,k}$. Thus lemma D.1 implies

$$P_4^* \odot (A\mathrm{J}) = (\mathrm{I} \odot A^2)P_4 - C_5$$
$$= (\mathrm{I} \odot A^2)(\mathrm{J} \odot A^4) - \mathrm{J} \odot ((\mathrm{I} \odot A^2)A(\mathrm{I} \odot A^2)A) + 2(\mathrm{I} \odot A^2)(\mathrm{J} \odot A^2) \tag{53}$$
$$- (\mathrm{I} \odot A^2)(\mathrm{I} \odot A^2)(\mathrm{J} \odot A^2) - (\mathrm{I} \odot A^2)(\mathrm{J} \odot A^2)(\mathrm{I} \odot A^2) - (\mathrm{I} \odot A^2)A(\mathrm{I} \odot A^3)$$
$$- (\mathrm{I} \odot A^2)(\mathrm{I} \odot A^3)A + 3(\mathrm{I} \odot A^2)(A \odot A^2) - A \odot A^4 + A \odot (A(\mathrm{I} \odot A^2)A)$$
$$+ (\mathrm{I} \odot A^2)(A \odot A^2) + (A \odot A^2)(\mathrm{I} \odot A^2) + A(\mathrm{I} \odot A^3) + (\mathrm{I} \odot A^3)A - 5A \odot A^2.$$

Fifteenth, we have $(AP_3^*)_{i,i} = \sum_k (C_{4f})_{i,k}$. Thus lemma D.1 implies

$$(\mathrm{J} \odot A^2) \odot (C_{4f}\mathrm{J}) = (\mathrm{I} \odot (AP_3))(\mathrm{J} \odot A^2) - (\mathrm{J} \odot A^2) \odot C_4$$
$$= \underbrace{(\mathrm{I} \odot (A(\mathrm{J} \odot A^3)))(\mathrm{J} \odot A^2)}_{=(\mathrm{I} \odot A^4)(\mathrm{J} \odot A^2)} - (\mathrm{I} \odot (A(\mathrm{I} \odot A^2)A))(\mathrm{J} \odot A^2)$$
$$- \underbrace{(\mathrm{I} \odot (A^2(\mathrm{I} \odot A^2)))(\mathrm{J} \odot A^2)}_{=(\mathrm{I} \odot A^2)(\mathrm{I} \odot A^2)(\mathrm{J} \odot A^2)} + (\mathrm{I} \odot A^2)(\mathrm{J} \odot A^2) - A^2 \odot A \odot A^3$$
$$+ \underbrace{(\mathrm{J} \odot A^2) \odot ((\mathrm{I} \odot A^2)A)}_{=(\mathrm{I} \odot A^2)(A \odot A^2)} + \underbrace{(\mathrm{J} \odot A^2) \odot (A(\mathrm{I} \odot A^2))}_{=(A \odot A^2)(\mathrm{I} \odot A^2)} - A^2 \odot A$$
$$= (\mathrm{I} \odot A^4)(\mathrm{J} \odot A^2) - (\mathrm{I} \odot (A(\mathrm{I} \odot A^2)A))(\mathrm{J} \odot A^2) \tag{54}$$
$$- (\mathrm{I} \odot A^2)(\mathrm{I} \odot A^2)(\mathrm{J} \odot A^2) + (\mathrm{I} \odot A^2)(\mathrm{J} \odot A^2) - A \odot A^2 \odot A^3$$
$$+ (\mathrm{I} \odot A^2)(A \odot A^2) + (A \odot A^2)(\mathrm{I} \odot A^2) - A \odot A^2.$$

And eventually, from the previous equality, we have

$$A \odot A^2 \odot P_3^* = A \odot A^2 \odot A^3 - (\mathrm{I} \odot A^2)(A \odot A^2) - (A \odot A^2)(\mathrm{I} \odot A^2) + A \odot A^2, \tag{55}$$

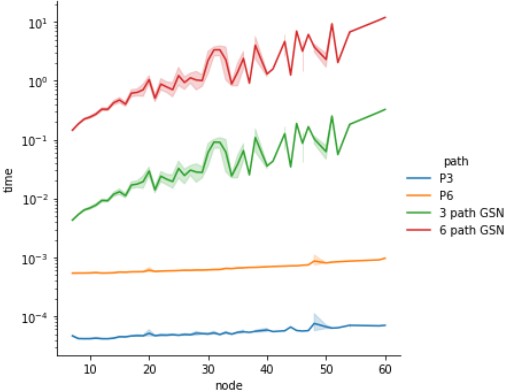

Figure 14: Time computation of GRL formulae and preconsumption of GSN (log scale) on IMDB-MULTI dataset for path of length 3 and 6.

and

$$C_{4f}A = \mathrm{J} \odot ((A \odot A^3)A) - \mathrm{J} \odot (A(\mathrm{I} \odot A^2)A) - (\mathrm{I} \odot A^2)(\mathrm{J} \odot A^2) + \mathrm{J} \odot A^2, \quad (56)$$

By removing the correct combination of $\mathrm{J} \odot (AP_5^*)$ and equations (51) to (56) to $P_6$, we obtain exactly $P_4^* + C_{5f} + 3\mathrm{J} \odot ((A \odot A^2)(\mathrm{J} \odot A^2)) + \mathrm{J} \odot A^2 \odot A^2 \odot A^2 + 3\mathrm{J} \odot ((A \odot A^2 \odot A^2)A) - 4\mathrm{J} \odot A^2 \odot A^2 - 8A \odot (A(A \odot A^2)) - 8A \odot ((A \odot A^2)A) - 4\mathrm{J} \odot ((A \odot A^2)A) - 3A \odot ((A \odot A^2)\mathrm{J}) + 17A \odot A^2 + 3\mathrm{J} \odot A^2$. It concludes the proof. $\qquad\square$

An alternative understanding of how $P_6^*$ computes the number of 5-paths connecting two nodes can be described as follows:

The expression $AP_6^*$ calculates, from a given node, a non-closed 5-path followed by a 1-path. This computation inherently includes non-closed 6-paths as well as 6-cycles. The 6-cycles are subsequently removed by the Hadamard multiplication with J, which zeroes out the diagonal elements. However, this operation also allows the possibility of traversing a 5-path and then returning to an intermediate node. To account for this and eliminate respectively paths returning to the fifth, fourth, third and second nodes of the 4-paths, we subtract the other terms in $P_6^*$.

Thanks to formula (26), we can derive the 7-cycle formula.

**Corollary D.5.1**
*The following formula, denoted as $C_{7f}$ computes the number of 7-cycles linking two nodes*

$$C_{6f} = A \odot (AP_5^*) - C_{4f} \odot ((A \odot A^2)\mathrm{J}) - A \odot (C_{5f}\mathrm{J}) - C_{5f} \odot (A\mathrm{J}) + 2C_{5f} \quad (57)$$

$$- (A \odot A^2) \odot (C_{4f}\mathrm{J}) + 4A \odot A^2 \odot P_3^* + 3A \odot ((A \odot A^2)(\mathrm{J} \odot A^2)) + A \odot A^2 \odot A^2 \odot A^2$$

$$+ 3A \odot ((A \odot A^2 \odot A^2)A) - 4A \odot A^2 \odot A^2 - 8A \odot (A(A \odot A^2)) - 12A \odot ((A \odot A^2)A)$$

$$- 3A \odot ((A \odot A^2)\mathrm{J}) + 20A \odot A^2.$$

In terms of time complexity, $P_6^*$ is more efficient than $P_6$. The ratio of time complexity of $P_6^*$ over $P_6$ is $\frac{4}{25}$. It is directly derived from the number of matrix multiplications in both formulas. Figure 7 shows the gain of complexity of $P_6^*$ and the ratio between the two formulas.

GRL improves the computation of path and cycle at edge-level for $l = 6$ by a factor 6.25.

We evaluated the computational time of $P_3^*$ and $P_6^*$ and the precomputation times of GSN on the IMDB-MULTI dataset, with the results shown in Figure 14. This analysis underscores the efficiency gains provided by the formulae.

## E  GRL APPLIED ON DIRECTED GRAPHS

In this section, we extend the application of GRL to derive matrix formulae for counting paths and cycles in directed graphs.

To adapt the grammar reduction approach proposed in Piquenot et al. (2024) for directed graphs, minor modifications are necessary. Specifically, the transpose operation becomes critical due to the asymmetry of the adjacency matrix in directed graphs. Despite this adjustment, the proof used to eliminate the $V_c$ variable remains valid. Consequently, the grammar $G_d$ provided to GRL for this task is defined as follows.

$$
\begin{aligned}
E &\to (E \odot M) \mid (NE) \mid (EN) \mid E^{\mathbf{T}} \mid A \mid \text{J} \\
N &\to (N \odot M) \mid (N \odot N) \mid \text{I} \\
M &\to (MM) \mid (EE) \mid M^{\mathbf{T}}
\end{aligned}
\tag{58}
$$

In addition to GRL, we explored search within the CFG using two alternative methods: MCTS without GramFormer, referred to as MC, and a completely random rule selection process, referred to as Rand. For both MC and Rand, we maintained the same rollout budget and maximum sentence length as those used for GRL. In the following, we denote the path matrix for directed graphs as $P_l^d$.

For $l = 2$, since the task can be resolved by a single short formula, GRL, MC, and Rand all successfully identified $P_2^d$.

$$
P_2^d = \text{J} \odot A^2.
$$

For $l = 3$, when GRL was initially applied to directed graphs, we aimed to identify a combination of four sentences to construct the formula. However, GRL determined that, similar to the case of undirected graphs, only two sentences were required. This resulted in the following simplified formula.

$$
P_3^d = \text{J} \odot (A(\text{J} \odot A^2)) - A \odot ((A \odot A^{\mathbf{T}})\text{J}).
$$

Both MC and Rand failed to converge when tasked with finding a combination of four sentences. However, after GRL revealed that only two sentences were necessary, we re-evaluated MC and Rand under this condition. In this case, both methods successfully converged and identified $P_3^d$.

For $l = 4$, only GRL successfully discovered a solution.

$$
\begin{aligned}
P_4^d = {}&\text{J} \odot (A(\text{J} \odot (A(\text{J} \odot A^2)))) - \text{J} \odot (A(A \odot ((A \odot A^{\mathbf{T}})\text{J}))) \\
&- \text{J} \odot ((A \odot ((A \odot A^{\mathbf{T}})\text{J}))A) - A \odot ((A \odot (A^2)^{\mathbf{T}})\text{J}) + 2 * A \odot A^{\mathbf{T}} \odot A^2.
\end{aligned}
$$

For $l = 5$, only GRL successfully discovered a solution.

$$
\begin{aligned}
P_5^d = {}&\text{J} \odot (A(\text{J} \odot (A(\text{J} \odot (A(\text{J} \odot A^2)))))) - \text{J} \odot (A(\text{J} \odot (A(A \odot ((A \odot A^{\mathbf{T}})\text{J}))))) \\
&- \text{J} \odot (A(\text{J} \odot ((A \odot ((A \odot A^{\mathbf{T}})\text{J}))A))) - \text{J} \odot (A(A \odot ((A \odot (A^2)^{\mathbf{T}})\text{J}))) \\
&- (\text{J} \odot A^2) \odot ((A^{\mathbf{T}} \odot A^2)\text{J}) - A \odot ((A \odot (A(\text{J} \odot A^2))^{\mathbf{T}})\text{J}) \\
&- ((A \odot A^{\mathbf{T}})\text{J}) \odot (\text{J} \odot (A(\text{J} \odot A^2))) - A \odot ((A \odot A^{\mathbf{T}})\text{J}) - A \odot A^{\mathbf{T}} \odot ((A \odot A^{\mathbf{T}})\text{J}) \\
&- A^{\mathbf{T}} \odot A^2 - 3A \odot (A \odot A^{\mathbf{T}})^2 + 2A \odot A^2 \odot +^{\mathbf{T}}A \odot A^{\mathbf{T}} \odot (A(\text{J} \odot A^2))(A^2) \\
&+ \text{J} \odot ((A \odot (A^2)^{\mathbf{T}})A) + 2\text{J} \odot ((A \odot A^{\mathbf{T}} \odot (A^2))A) \\
&+ \text{J} \odot ((A \odot A^{\mathbf{T}})(\text{J} \odot A^2)) + ((A \odot A^{\mathbf{T}})\text{J}) \odot (A \odot ((A \odot A^{\mathbf{T}})\text{J})) \\
&+ A \odot (((A \odot A^{\mathbf{T}}) \odot ((A \odot A^{\mathbf{T}})\text{J}))\text{J}) + 2 * \text{J} \odot (A(A \odot A^{\mathbf{T}} \odot A^2)).
\end{aligned}
$$

To the best of our knowledge, the matrix formulae derived for counting paths of lengths 2 to 5 are novel. This highlights the capability of GRL to generate new and meaningful formulae. Furthermore, when assuming that the adjacency matrix $A$ is symmetric, the formulae discovered by GRL for directed graphs align perfectly with those identified for undirected graphs. This alignment leads us to conjecture that GRL has identified optimal formulae, at least in the undirected case.

# F  ALGORITHMS

This section provides the pseudo code of algorithm of section 2 through 4.

---

**Algorithm 1:** PDA sentence generation for a given grammar $G$.

---

**Input:**
    The PDA $(Q, \Sigma, \Gamma, \delta, q_0, Z, F)$ derived from a given CFG $G$.
**Output:**
    A sentence $w \in L(G)$ where each element of $w$ is in $\Sigma$.
**Initialisation:**
    $S \leftarrow [Z]$ # Initialisation of the stack
    $w \leftarrow [\,]$ # Initialisation of the written sentence
**while** $S \neq [\,]$ **do**
    $c \leftarrow pop(S)$
    **if** $c \in \Sigma$ **then**
        Append $c$ to $w$ # Write the terminal symbol $c$
    **else**
        Choose $t \in \delta(q_0, \varepsilon, c)$ # Choose a transposition for the variable $c$
        $push(S, t)$ # Concatenate the rule $t$ to the stack
**Return:** $w$

---

**Algorithm 2:** Definition of the algorithm $read$, that return the first variable token in input $I$ and its position if $I$ contains variable token

---

**Input:**
    The set of token $T := \{T_v, T_r, T_t\}$ derived from a given CFG $G$
    The Input $I$, concatenation of $w \in \Sigma^*$ the written terminal symbols with the stack $S \in \Gamma^*$
**Output:**
    $b$ a boolean that indicate whether $I$ contains elements of the variables token set $T_v$
    $v \in T_v$ a variable token, None if $b$ is False
    $pos$ the position of $v$ in $I$, None if $b$ is False
**Initialisation:**
    $b \leftarrow False$ $v \leftarrow None$ $pos \leftarrow None$
**for** $c, i \in enumerate(I)$ **do**
    **if** $c \in T_v$ **then**
        $b \leftarrow True$
        $v \leftarrow c$
        $pos \leftarrow i$
        Break
**Return:** $b, v, pos$

---

# G  GRL PARAMETER FOR THE PATH COUNTING PROBLEM

To ensure reproducibility of our results, we provide detailed specifications of the hyperparameters used in our experiments:

MCTS PARAMETERS

- **Exploration Parameter ($c$):** The exploration-exploitation trade-off parameter in the UCT formula is set to $c = 10$.

- **Rollouts per Node:** Each node is evaluated using 10000 rollouts, with a maximum trajectory length of 20 steps.

---

**Algorithm 3:** Prediction algorithm of the gramformer model $M = (encoder, decoder)$.

---

**Input:**

    A variable token $v \in T_v$ of the set of token $T := \{T_v, T_r, T_t\}$ derived from a given CFG $G$

    A variable mask $M_v$ corresponding to the variable token $v$

    The Input $I$, concatenation of $w \in \Sigma^*$ the written terminal symbols with the stack $S \in \Gamma^*$

**Output:**

    $policy$ the learned distribution of possible rules selection form the variable corresponding
        to $v$

    optional:$value$ the learned empirical value

$memory \leftarrow encoder(v)$
$latent \leftarrow decoder(memory, I)$
$value \leftarrow MLP(latent, 1)$ # Optional
$pol \leftarrow MLP(latent, nbtoken)$
$policy \leftarrow softmax(pol + M_v)$ # probability distribution of the possible transposition of token
  variable $v$
**Return:** $policy$,($value$ # Optional)

---

**Algorithm 4:** $Replace$ the variable of $I$ at position $pos$ by the list of variable and/or terminal tokens corresponding to the rule token at indices $decision$

---

**Input:**

    The Input $I$, concatenation of $w \in \Sigma^*$ the written terminal symbols with the stack $S \in \Gamma^*$

    The position of the variable token to replace $pos$

    The indice $decision$ of the selected rule token

**Output:**

    A new $I$ where, the variable at position $I$ as been replaced by the list of variable and/or
    terminal tokens corresponding to the rule token at indices $decision$

$sb \leftarrow I[:pos]$
$sf \leftarrow I[pos+1:]$
$si \leftarrow tokens(decision)$
**Return:** $concatenate(sb, si, sf)$

---

**Algorithm 5:** Gramformer sentence generation for a given grammar $G$.

---

**Input:**

    The transformer model $M$, the set of token $T := \{T_v, T_r, T_t\}$ and the dictionary of variable
    mask $M_v$ derived from a given CFG $G$.

**Output:**

    A sentence $w \in L(G)$ where each element of $w$ is in $\Sigma$.

**Initialisation:**

    $I \leftarrow [Z]$ # Initialisation of the input

    $b, v, pos \leftarrow read(I)$ # read the input

**while** $b$ **do**

    $policy \leftarrow M(v, M_v[v], I)$ # Distribution proposed by the transformer model

    $decision \leftarrow argmax(policy)$

    $I \leftarrow replace(I, pos, decision)$

    $b, v, pos \leftarrow read(I)$

**Return:** $I$

---

---

**Algorithm 6:** GRL algorithm one agent acting phase

---

**Input:**
- A $MCTS$ defined to search within a CFG $G$
- $nbwords$, the number of sentences to generate
- A fixed Gramformer $M$
- A $buffer$

**Output:**
- A $buffer$ that contains empirical policy and value of the tree explored during MCTS for the selected nodes of this MCTS.

**Initialisation:**
- $I \leftarrow [Z] * nbwords$ # Initialisation of the input
- $b, v, pos \leftarrow read(I)$ # read the input
- $tree \leftarrow initMCTS(nbwords)$
- $buffer \leftarrow [I]$

**while** $b$ **do**
- $root \leftarrow tree(I)$
- $decision, tree \leftarrow MCTS(root, M)$
- $I \leftarrow replace(I, pos, decision)$
- $b, v, pos \leftarrow read(I)$
- Append $I$ to $buffer$

**Return:** fill($buffer, tree$)

---

**Algorithm 7:** MCTS algorithm

---

**Input:**
- The gramformer model $M$, and $root$, a node of the tree.

**Output:**
- A $child$ of the root node guided by the MCTS heuristic.

**for** $i=1$ to N (Number of simulations) **do**
- $leaf \leftarrow selection(root, M)$
- $child \leftarrow Expansion(leaf)$
- $reward \leftarrow simulation(child)$
- $Backpropagation(child, reward)$

**Return:** $Bestchild(root, M)$

---

**Algorithm 8:** MCTS selection step

---

**Input:**
- The gramformer model $M$ and a $node$ of the tree

**Output:**
- A descendant node of the input $node$ guided by the MCTS heuristic.

**while** $node$ *is fully expanded and not leaf* **do**
- $node \leftarrow Bestchild(node, M)$

**Return:** $node$

---

**Algorithm 9:** MCTS expansion step

---

**Input:**
- a $node$ of the tree

**Output:**
- A descendant node of the input $node$ or the input $node$.

**if** $node$ *is not leaf and not fully expanded* **then**
- $node \leftarrow RandomUnvisitedChild(node)$

**Return:** $node$

---

---

**Algorithm 10:** MCTS simulation step

---

**Input:**
└ a $node$ of the tree
**Output:**
└ A reward.
**while** $node\ is\ not\ leaf$ **do**
  └ $node \leftarrow RandomAction(node)$
**Return:** $Evaluate(node)$

---

**Algorithm 11:** MCTS backpropagation step

---

**Input:**
└ a $node$ of the tree, a $reward$.
**while** $node\ is\ not\ root$ **do**
  │ Add 1 to the visit count of $node$
  │ Add $reward$ to the reward count of the $node$
  └ $node \leftarrow$ parent of $node$

---

**Algorithm 12:** MCTS bestchild algorithm

---

**Input:**
└ The gramformer model $M$ and a $node$ of the tree
**Output:**
└ The best child of $node$.
$policy, - \leftarrow M(node)$
$N \leftarrow$ visits count of $node$
$res \leftarrow \{\ \}$
**for** $child\ of\ node$ **do**
  │ $-, reward \leftarrow M(child)$
  │ $N_c \leftarrow$ visits count of $child$
  │ $v \leftarrow M(child)$
  │ $Q \leftarrow$ value of $child$ / visits count of $child$
  └ $res[child] \leftarrow \alpha Q + (1 - \alpha)v + c \times policy(child) \times \frac{\sqrt{\sum_c N_c}}{1+N}$
**Return:** $argmax(res)$

---

**Algorithm 13:** GRL learning phase algorithm

---

**Input:**
│ The gramformer model $M$ and a $buffer$ containing policy and value for given nodes of
└ former trees.
**Output:**
└ The updated gramformer $M$
$optimiser \leftarrow ADAM(M, lr)$
**for** $i=1\ to\ N\ (number\ of\ epoch)$ **do**
  │ **for** $node\ in\ buffer$ **do**
    │ $policy, reward \leftarrow M(node)$
    │ $policyLoss \leftarrow KLLoss(policy, \text{policy of } node)$
    │ $valueLoss \leftarrow HubertLoss(reward, \text{value of } node)$
    │ $Loss \leftarrow policyLoss + valueLoss$
    │ $backward(Loss)$
    └ step of $optimser$
**Return:** $M$

---

TRANSFORMER ARCHITECTURE (GRAMFORMER)

- **Number of Layers:** 4 transformer layers.
- **Hidden Dimension:** 256.
- **Number of Attention Heads:** 8.
- **Feedforward Dimension:** 512.
- **Positional Encoding:** Learned positional embeddings for sentences up to 100 tokens for each sentences.

OPTIMIZER AND TRAINING PARAMETERS

- **Optimizer:** Adam optimizer with $\beta_1 = 0.9$, $\beta_2 = 0.999$, and $\epsilon = 10^{-8}$.
- **Learning Rate:** A learning rate of $10^{-4}$ was used.
- **Batch Size:** 128 sentences per batch.

**Resource consumption**   GRL, like other MCTS based algorithm, suffers from significant resource consumption. For the 5-path and 6-path counting problems, it took weeks to converge. Even with substantial CPU resources (124 AMD EPYC 9654) for the acting and GPU (4 NVIDIA A100) resources for the learning, the process remains extremely time-consuming.

These hyperparameter choices are consistent across all experiments unless stated otherwise. Further details specific to individual tasks and datasets are provided in the respective sections of the Appendix.

