# OpenReview forum: "Grammar Reinforcement Learning: path and cycle counting in graphs with a Context-Free Grammar and Transformer approach"
_ICLR.cc/2025/Conference — ICLR 2025 Poster_

### Official Review · Reviewer_LKha · 2024-10-27

**Soundness:** 2
**Presentation:** 2
**Contribution:** 2
**Rating:** 6
**Confidence:** 3

**Summary:**

The paper introduces a new approach to efficiently count paths and cycles in graphs. GRL uses a pushdown automaton (PDA) approach to generate and optimize mathematical formulas for path and cycle counting, addressing computational challenges in fields such as network analysis, biology, and social sciences. By framing path/cycle counting as a CFG-constrained search problem, GRL discovers new matrix-based formulas, improving computational efficiency by two to six times over current methods. Key contributions include a generic framework for generating efficient formulas within a CFG, the development of Gramformer to learn policies and values within a PDA model, and the identification of novel, efficient counting formulas for graph substructures.

**Strengths:**

- Great literature work: This paper references existing literature, which demonstrates an awareness of previous work in the area and situates the study within the broader research context.
- Full details of the proposed method with detailed theoretical justification.

**Weaknesses:**

- The focus of this work is on re-solving a problem that requires a polynomial-time algorithm, as indicated in Line 99: "As mentioned in Section 1, path/cycle counting has been extensively tackled in the literature." This context renders the proposed methods less impactful. I recommend that the authors provide a more compelling rationale for why this issue is considered challenging within the fields of network analysis, computer science, biology, and social sciences.

- There are no experiments conducted on datasets from network analysis, computer science, biology, or social sciences. Given that the results in Figure 7 indicate an algorithm running in 0.2 seconds, which is already considered "very fast," it is unclear what specific challenge is being addressed in this figure.

- The mathematical notation used throughout the paper tends to create confusion rather than enhance the reader's understanding of the problem.

**Questions:**

- Could you help to answer my concern on the first point of weakness?
- Could you clarify the purpose of the cubic shapes in Figures 3, 4, and 5? Are they intended to convey a particular meaning beyond their function as symbols?

---

> ### Author Response · Authors · 2024-11-22
>
> # Response to Reviewer Comments
>
> We thank the reviewer for their thorough and constructive feedback. We are pleased to see that they appreciate the contributions of our article. Below, we address the specific points raised in the review.
>
> ---
>
> ## **1. First Weakness: Focus on a Polynomial-Time Problem**
>
> Regarding the first weakness raised by the reviewer, we would like to emphasise that the focus of our work goes beyond “re-solving a problem that requires a polynomial-time algorithm.” The main contribution of our work lies in the development of an **algorithm design framework** capable of generating new algorithms with desirable properties, provided the problem can be framed as a CFG-constrained language generation problem.
>
> In this regard, our approach is conceptually similar to the methodology in [fawsi], though applied in a different context, which necessitated contributions such as GramFormer. In our paper, the task of designing path/cycle extraction formulae serves as a use case to demonstrate the potential of this framework. By leveraging existing solutions (such as Voropaev formulae) as baselines, our goal is to show that a machine learning framework can design solutions of human-level quality or even exceed them for this use case.
>
> To further demonstrate the genericity of the framework, we have added a new use case during the rebuttal period. This use case involves counting paths in **directed simple graphs**, requiring modifications to the grammar to account for the transposition operation (see Appendix E for details). Using this adapted grammar, GRL successfully derived formulae for $P_2, P_3, P_4,$ and $P_5$, which, to the best of our knowledge, do not exist in the literature. Interestingly, these formulae match their counterparts for undirected graphs, providing evidence supporting the conjecture that the GRL-discovered formulae may be optimal in both settings.
>
> Additionally, in the context of this new application, we conducted experiments comparing GRL with alternative approaches, such as using MCTS without the assistance of GramFormer and a purely random exploration of the grammar tree. These results, detailed in Appendix E, underline the impact and necessity of our GramFormer component.
>
> ---
>
> ## **2. Second Weakness: Limited Experimental Applications**
>
> The second weakness mentioned by the reviewer concerns the limited application domains in our experiments. We fully agree that the experimental section of the paper is not extensive. This limitation stems from the theoretical guarantees provided by the new counting formulae, which ensure that the observed time savings are domain-independent.
>
> However, we acknowledge that the time savings demonstrated for our initial use case may not seem groundbreaking. As stated earlier, the goal of this paper is not to show domain-specific breakthroughs but to present a **generic and powerful framework**. The new results on directed graphs (see Appendix E) provide a more compelling illustration of GRL's potential, especially as they generate previously unknown formulae.
>
> To reinforce the experimental evaluation, we have compared the computation time required by a classical algorithm for computing paths and cycles with the use of our formulae. Specifically, we take as a baseline the preprocessing steps of GSN, a GNN framework proposed by [1], which incorporates path and cycle counts at the node or edge level into feature representations. The results, detailed in Appendix D, demonstrate that our formulae significantly outperform the baseline in terms of computational efficiency. This comparison highlights the practical benefits of our contributions, particularly in scenarios where efficiency is critical.
>
>
>
>
> ---
>
> ## **3. Third Weakness: Confusing Notations**
>
> Regarding the third weakness raised by the reviewer, we have revised the manuscript to address concerns about the mathematical notations, incorporating suggestions from this reviewer and others. These changes are highlighted in red in the new version of the paper.
>
> ---
>
> ## **4. Second Question: Purpose of the Cubic Shapes in Figures 3, 4, and 5**
>
> Regarding the second question, the cubic shapes in Figures 3, 4, and 5 do not carry any specific meaning beyond their function as symbols. These shapes follow a common visual style used in papers concerning transformers and related architectures. Their inclusion is primarily aesthetic, aiming to provide a clean and familiar representation.
>
> ---
>
> ## **Conclusion**
>
> We thank the reviewer once again for their valuable feedback. The comments have helped us improve the clarity and scope of our work, and we hope that the new additions, particularly the directed graph use case, better highlight the potential of GRL as a generic algorithm design framework.

---

> > ### Author Response · Authors · 2024-11-22
> >
> > ## **References**
> >
> > [1] Bouritsas, G., Frasca, F., Zafeiriou, S., & Bronstein, M. M. (2022). Improving graph neural network expressivity via subgraph isomorphism counting. IEEE Transactions on Pattern Analysis and Machine Intelligence, 45(1), 657-668.

---

> > ### Comment · Reviewer_LKha · 2024-11-29
> > **reply**
> >
> > Thanks for addressing my concern. I'm happy to improve my evaluation score.

---

### Official Review · Reviewer_5jnX · 2024-11-03

**Soundness:** 2
**Presentation:** 3
**Contribution:** 2
**Rating:** 8
**Confidence:** 4

**Summary:**

This paper considers the problem of path and cycle counting in graphs. It aims to do so efficiently using formulae that perform a set of pre-defined matrix operations, specified as a context-free grammar. The authors frame this task as a combinatorial optimization problem and propose a reinforcement learning method to solve it. The method is made up of a Monte Carlo Tree Search with neural networks for function approximation. Particularly, the authors propose a transformer architecture that can process tokens from the grammar. The authors demonstrate that their method discovers more efficient formulae for path and cycle counting than those that were previously known.

**Strengths:**

S1. The authors treat an important problem with an interesting and novel methodology, and use it to discover demonstrably better path and cycle search algorithms.

S2. The paper is fairly well-written and the technique appears sound.

**Weaknesses:**

W1. The biggest weakness of the work is its very limited evaluation. The paper proposes a very complex methodology for operating in this discrete search space, but does not compare it with meaningful baselines. It is therefore not possible to determine whether the proposed method is indeed a better means of navigating this search space than other more standard methods, especially given the very large reported computational cost. At the very least, in my opinion, the paper should include empirical comparisons with:

- Online Monte Carlo Tree Search without any function approximation (to demonstrate the benefit of using the transformer within the search);
- The learned transformer policy at convergence;
- A classic metaheuristic such as simulated annealing;
- A simple random search that samples from the grammar and is given the same amount of rollouts as MCTS.

W2. Another weakness of the work is that the method does not generate algorithms that are provably correct. Indeed, the authors have to resort to proving the correctness of the algorithms themselves, which they do. This stems from the fact that, to evaluate a given formula, the method needs to compare the output over a limited set of graphs with generated ground truth values. It could be the case that a formula produces the correct outputs for this set of examples while missing some edge cases. This is a limitation that should be acknowledged and discussed.

W3. The literature review omits many related works on reinforcement learning for combinatorial optimization over graphs. I would suggest including at least [1], one of the first recent works to treat this type of problem with RL, and which also uses token-like sequences; [2], a work that made substantial leaps in RL for combinatorial optimization in terms of performance and scalability; as well as other recent works in this space (see [3] for a survey).

[1] Vinyals, O., Fortunato, M., & Jaitly, N. (2015). Pointer networks. Advances in neural information processing systems, 28.

[2] Khalil, E., Dai, H., Zhang, Y., Dilkina, B., & Song, L. (2017). Learning combinatorial optimization algorithms over graphs. Advances in neural information processing systems, 30.

[3] Darvariu, V.-A., Hailes, S., & Musolesi, M. (2024). Graph Reinforcement Learning for Combinatorial Optimization: A Survey and Unifying Perspective. Transactions on Machine Learning Research.

**Questions:**

C1. Given the usage of a reinforcement learning approach, the Markov Decision Process (i.e., the state/action/transitions/rewards) should be clearly specified in mathematical terms. The current textual description is ambiguous.

C2. The parameters used in your model (e.g., exploration parameter for MCTS, architectural parameters and optimizer / learning rate etc. for the transformer) should be clearly specified in an Appendix. The reproducibility is limited otherwise.

C3. There are a few clashes in notation because of trying to unite RL and CFG/DFA notations. For example, Q is used to refer to both the set of states of the automaton as well as the Q-value in RL. I'd suggest checking that each symbol is used consistently.

C4. I would suggest discussing whether and how the approach could apply to directed graphs as well (it seems undirected graphs are assumed).

C5. Could you also comment on what would be needed to generalise the approach to cycles of arbitrary length (more than 7)? Is it simply a matter of applying your method more computational power, or is the approach limited in this sense to predefined path lengths?

C6. Typos: "ouputs" (Fig 6 caption)

---

> ### Author Response · Authors · 2024-11-22
>
> # Response to Reviewer Comments
>
> We thank the reviewer for its thorough and constructive review, which has enabled us to enhance our paper by incorporating additional experiments, such as comparison with MCTS and random selection and formulae on directed graphs. In the following, we answer the reviewer's concerns and questions.
>
> ---
>
> ## **1. First Weakness: Limited evaluation and lack of meaningful baselines**
>
> To address this point, we conducted an evaluation of MCTS without the assistance of GramFormer, as well as a fully random selection within the grammar tree, using the same rollout budget and maximum sentence length as GRL. These results, detailed in Appendix E, underline the impact and necessity of our GramFormer component. As noted in Section 6.1 of [3], simulated annealing is unlikely to be effective for tasks involving long-horizon decision-making. Since searching within a CFG inherently involves a long-horizon decision, this approach is unlikely to yield satisfactory results in this context. Regarding the learned policies, since they are utilized throughout the entire MCTS process in GRL, we did not conduct a separate comparison.
>
> ---
>
> ## **2. Second Weakness: Lack of provable correctness**
>
> If all possible walks of length $l$ are represented within the graphs in the dataset used by GRL, the outputs of GRL inherently confirm the correctness of the derived formulae. Specifically, when GRL identifies a set of sentences with maximal value, this ensures that the set accurately counts the $l$-paths or cycles. We are convinced that the set of graphs on which the linear combinations of sentences are evaluated contains all walks of length less than $l$. Moreover, the successful derivation of equivalence proofs demonstrates that GRL's results are not only correct but also fully interpretable and verifiable by humans. These proofs further validate that the values computed by GRL confirm the correctness of the formulae and our assumption on the completeness of the graph dataset.
>
> ---
>
> ## **3. Third Weakness: Omissions in the literature review**
>
> We thank the reviewer for their insightful feedback, which led us to identify and include relevant references in the bibliography, enriching the paper.
>
> ---
>
> ## **4. First Question (C1) : Specification of the Markov Decision Process (MDP)**
>
> We have clarified several points in the paper to better connect the standard MCTS notation, as used in works like [1][2], to the reinforcement learning framework of Markov Decision Processes. Specifically, we have added the explanation:
> “From a reinforcement learning perspective, $ \frac{Q(I, r)}{N(I, r)} $ represents the expected return over all possible trajectories that originate from state $ I $ and take action $ r $.”
>
> Additionally, we have clarified that the UCB equation used to select a rule/action $ r $ from a node/state $ I $ corresponds to the transition step in an MDP. Finally, we have incorporated the following statement into the paper:
>
> “Mathematically, during the backward pass, the value of the trajectory leading to the currently evaluated leaf node is updated using the following equation.”
>
> \begin{align*}
> v(I_{\mathrm{parent}},r_{\mathrm{parent}})^{(k+1)} = v(I_{\mathrm{parent}},r_{\mathrm{parent}})^{(k)} + \mu v(I_{\mathrm{child}},r_{\mathrm{child}})^{(k+1)} - P_{r_{\mathrm{parent}}}
> \end{align*}
>
> ---
>
> ## **5. Second Question (C2) : Model parameters**
>
> We have included the hyperparameters used in our experiments in Appendix G of the paper for clarity and reproducibility.
>
> ---
>
> ## **6. Third Question (C3) : Notational consistency**
>
> To avoid any confusion between the set of PDA states and the $ Q $-value in our formulation, we have revised the notation in the paper to represent the set of PDA states as $ \mathcal{Q} $. We thank the reviewer for highlighting this point.
>
> ---
>
> ## **7. Fourth Question (C4) : Applicability to directed graphs**
>
> Motivated by this reviewer question, we have added a new use case during the rebuttal period, extending GRL's application to directed simple graphs. This extension required modifying the grammar to account for the transposition operation, as detailed in Appendix E. Using this adapted grammar, GRL successfully derived formulas for $P_2$, $P_3$, $P_4$, and $P_5$, which, to the best of our knowledge, are novel and not present in the literature. Interestingly, these formulas align with their counterparts for undirected graphs, providing evidence supporting the conjecture that the GRL-discovered formulas may be optimal at least in undirected cases. We would like to thank the reviewer for this suggestion.

---

> > ### Author Response · Authors · 2024-11-22
> >
> > ## **8. Fifth Question (C5) : Generalization to cycles of arbitrary length**
> >
> > As discussed in the paper, our limitation in deriving path formulas beyond length 6 stems from the expressive power of 3-WL. It has been established that 3-WL cannot count cycles of length 8 at the graph level, and the path/cycle formulas in [3] further imply that 3-WL cannot count paths of length 7. Since $G_3$ is designed to match the expressiveness of 3-WL, this limitation naturally applies to our approach as well. Extending beyond this constraint requires greater expressive power. To address this, we are actively working on developing $k$-WL CFGs to enable GRL to operate on more expressive grammars. Our aim is to derive formulas with asymptotic complexity improvements in the future.
> >
> > ---
> >
> > ## **9. Sixth Question (C6) : Typos**
> >
> > Thank you for the typos which have been corrected.
> >
> > ---
> >
> > ## **Conclusion**
> >
> > We sincerely thank the reviewer for their detailed and constructive feedback, which has helped us identify key areas for improvement and refinement. Through this rebuttal, we have incorporated additional experiments, addressed important methodological clarifications, and extended our framework to new use cases, such as directed graphs. These enhancements have strengthened our paper and highlighted the versatility of our approach.
> >
> > We hope that this rebuttal addresses the majority of the concerns raised in the review.
> >
> > ## References
> >
> > [1] Silver, D., Hubert, T., Schrittwieser, J., Antonoglou, I., Lai, M., Guez, A., ... & Hassabis, D. (2018). A general reinforcement learning algorithm that masters chess, shogi, and Go through self-play. Science, 362(6419), 1140-1144.
> > [2] Fawzi, A., Balog, M., Huang, A., Hubert, T., Romera-Paredes, B., Barekatain, M., ... & Kohli, P. (2022). Discovering faster matrix multiplication algorithms with reinforcement learning. Nature, 610(7930), 47-53.
> > [3] Voropaev, A. N., & (2009)Perepechko, S. N. The number of fixed length cycles in an undirected graph. Explicit formulae in case of small lengths. Mathematical Modeling and Computational Physics (MMCP2009), 148.

---

> ### Comment · Reviewer_5jnX · 2024-11-26
>
> Thanks for engaging with my comments. A few further replies (numbered according to the rebuttal provided):
>
> ### 1. First Weakness: Limited evaluation and lack of meaningful baselines
> Thanks for the additional results. While these do provide some signal, I think it would be much more solid methodologically to compare the actual rewards received by the agents across multiple runs, as opposed to reporting the "outcome" in terms of the discovered formulae across one repetition.
>
> ### 2. Second Weakness: Lack of provable correctness
> I see your point -- but while you can enumerate all walks for a particular graph, you cannot hope to do so for all graphs. There is still the possibility that a formula that is correct for one graph (or a finite number of graphs) does not hold in general. And, by definition, this set needs to be finite. I still think this work can be a useful tool for discovering formulae, but the method itself is not guarranteed to find correct formulae by default.
>
> ### 4. First Question (C1) : Specification of the Markov Decision Process (MDP)
> I think you have misunderstood my point. For an example, see Section 4.1 of the Khalil et al. paper mentioned above. The details you have added are about the MCTS mechanics and are not relevant in my opinion. The definition of the MDP (what I suggested to add) is independent from the algorithm you choose to solve it (which happens to be MCTS).
>
> I have updated my score to an 8: while the paper does still have shortcomings, I think the contributions outweigh them.

---

> > ### Author Response · Authors · 2024-11-29
> >
> > We would like to once again thank the reviewer for their thorough analysis of our article and for revising their assessment upwards.  Concerning the second point, we are committed to working on a formal proof to validate our hypothesis. Lastly, regarding the final point, we apologize for not fully understanding the reviewer’s request and will address this in the final version of the paper, should it be accepted.

---

### Official Review · Reviewer_ZBFH · 2024-11-04

**Soundness:** 3
**Presentation:** 3
**Contribution:** 2
**Rating:** 8
**Confidence:** 3

**Summary:**

The paper presents a reinforcement learning setting for the generation of formulae that allow for efficient counting paths and cycles up to length six in directed graphs by implementing a transformer-guided Monte Carlo tree search on a generative syntax tree of a suitable context-free grammar.

While the application of efficiently counting cycles is very intuitive, and the method of transformer-guided MCTS on formal grammar is very interesting, the paper misses any discussion of related work concerning RL methods of deep neural network-guided MCTS or MCTS on formal grammar. The paper would also heavily benefit from discussing other use cases of transformer-guided MCTS on formal grammar for GRL such that the paper could have focused more on the method of GRL rather than a specific use case only. The explanations and proofs also need clarity and seem incomplete or partly wrong.

**Strengths:**

- The paper introduced an exciting setup of transformer-guided MCTS on context-free grammar in a reinforcement learning setting for generating specific words of the corresponding context-free language.
- The chosen application of efficient counting cycles is quite versatile in real-world scenarios.

**Weaknesses:**

- The paper does not mention any other work related to the used/introduced machine learning method of grammar/language/llm-guided MCTS (the cited work is only about the chosen application of efficiently counting paths/cycles in graphs).
- The paper could include other GRL applications to strengthen the introduced method's flexibility.
- The relevance of counting cycles could be better illustrated with a few examples next to the literature reference.
- Some explanations need coherence and clarity, especially the usage of PDAs in the paper, which could be more accurate or even omitted. (see Questions)
- At least two proofs must be completed or corrected, and a proper list of necessary conditions/definitions could improve readability in most cases. (see Questions)

**Questions:**

Regarding Proofs:
- In the proof of theorem 3.1/A.1 it says that $(N \times \mathrm{J} \times \mathtt{diag}(w)) \cdot \mathrm{I} = \mathtt{diag}((N \cdot \mathrm{J}) \times w)$ but if $n=2$, $N = \begin{bmatrix} a & b \\\\ c & d \end{bmatrix}$, $w = \begin{bmatrix} x \\\\ y \end{bmatrix}$, we have $(\begin{bmatrix} a & b \\\\ c & d \end{bmatrix} \times \begin{bmatrix} 0 & 1 \\\\ 1 & 0 \end{bmatrix} \times \begin{bmatrix} x & 0 \\\\ 0 & y \end{bmatrix}) \cdot \begin{bmatrix} 1 & 0 \\\\ 0 & 1 \end{bmatrix} = (\begin{bmatrix} b & a \\\\ d & c \end{bmatrix} \times \begin{bmatrix} x & 0 \\\\ 0 & y \end{bmatrix}) \cdot \begin{bmatrix} 1 & 0 \\\\ 0 & 1 \end{bmatrix} = \begin{bmatrix} bx & 0 \\\\ 0 & cy \end{bmatrix}$ and $(\begin{bmatrix} a & b \\\\ c & d \end{bmatrix} \cdot \begin{bmatrix} 0 & 1 \\\\ 1 & 0 \end{bmatrix}) \times \begin{bmatrix} x \\\\ y \end{bmatrix} = \begin{bmatrix} 0 & b \\\\ c & 0 \end{bmatrix} \times \begin{bmatrix} x \\\\ y \end{bmatrix} = \begin{bmatrix} by \\\\ cx \end{bmatrix}$, hence, $\mathtt{diag}(\begin{bmatrix} by \\\\ cx \end{bmatrix}) = \begin{bmatrix} by & 0 \\\\ 0 & cx \end{bmatrix}$. Therefore, I do not yet see how the induction can work.
- In the proof of theorem D.1, it is used that $\mathrm{J} \cdot (A\times(\mathrm{I} \cdot (A \times A))) = A\times(\mathrm{I} \cdot (A \times A))$ but if $A=\begin{bmatrix} 1 & 1 \\\\ 1 & 1 \end{bmatrix}$, we have $A \times A = 2A, \mathrm{I} \cdot A = \mathrm{I}, \mathrm{J} \cdot A = \mathrm{J}$ and hence $A\times(\mathrm{I} \cdot (A \times A)) = A \times 2\mathrm{I} = 2A  \neq 2\mathrm{J} = \mathrm{J} \cdot (A\times(\mathrm{I} \cdot (A \times A)))$.

Regarding PDA:
- The usage of nondeterministic PDA is a bit confusing as PDAs are usually language acceptors rather than language generators. Still, of course, this is technically just a relabelling of the input as output. This happens in the text without clarification.
- There is also an inconsistent use of the terms 'transition relation' vs. 'transition function'.
- The PDAs could be omitted entirely using generative syntax trees induced by context-free grammar as there is no build-up on automata theory. For Gramformers, this would mean defining a variable token for every nonterminal character, a rule token for every production, and a terminal token for every terminal character. This would make the approach much easier to follow.

---

> ### Author Response · Authors · 2024-11-21
>
> We thank the reviewer for his thorough evaluation of our work, which has allowed us to identify an error in our proofs and to improve the paper by expanding the bibliography and including new experiments that highlight the potential and relevance of GRL.
> Despite the criticism, we are very pleased that the reviewer found the GRL setup exciting.
>
> Below, we address the specific points raised in the review:
>
> To address the first weakness outlined by the reviewer, we add a paragraph in Section 1 of the paper that presents contemporain works related to grammar search in different domains. “The use of context-free grammars (CFGs) for search has gained considerable attention in the literature. In robotics, [1] demonstrated that CFGs can effectively describe robotic structures, enabling the design and adaptation of robots for diverse tasks to be formulated as a search problem within a CFG. Similarly, in the field of AutoML, the application of CFG-based search has been explored ([2],[3]). In particular, [4] proposed a grammar specifically designed for an AutoML task, where searching within this grammar enables the discovery of the optimal model for a given task.”
>
> To address the second weakness and in response to Reviewer 5jnX's question C4, we have extended the application of GRL to the same task of counting paths, but for directed simple graphs. In this context, the grammar has to be modified in order to include the transposition operation (please see appendix E ). With this new grammar as input to the framework, GRL successfully derived formulae for  $P_2$, $P_3$, $P_4$, and $P_5$​, which, to the best of our knowledge, do not exist in the literature. Interestingly, these formulae obtained for directed graphs match their counterparts for undirected graphs, providing evidence for a conjecture about the optimality of the GRL-discovered formulae in both settings. Please note that in the context of this new application, we have compared the application of GRL with an evaluation of MCTS without the assistance of GramFormer, as well as a fully random selection within the grammar tree. Obtained results highlight the impact of our gramformer proposal (please see Appendix E of the new version of the paper).
>
> To address the third weakness  regarding the significance of path and cycle counting, we have revised the first paragraph of our introduction to reference the foundational work that introduced GSN [5], a GNN framework that incorporates path and cycle counts at the node or edge level into feature representations. This methodology has been shown to substantially enhance the expressive power of GNNs. The authors of GSN emphasize the significant computational cost associated with precomputing paths and cycles, highlighting the necessity of developing more efficient algorithms for path and cycle counting. In this context, we have compared the computation time required by the GSN preprocessing step with the use of our formulae. The results clearly emphasize the interest of our contributions (see the figure below comparing execution times).
>
> Similarly, the expressive power of graph transformers  [6] largely depends on the positional encoding of graphs. Incorporating counts of substructures such as paths and cycles as part of the positional encoding could improve their effectiveness. Our work is a contribution in this direction.
>
>
> The fourth and fifth weaknesses identified by the reviewer are addressed in our response to the questions.
>
>
> **Regarding Proof A.1:**
> The reviewer is absolutely right. There was indeed an error in the proof : $\mathrm{diag}(w)$ and $\mathrm{J}$ have been interchanged. The correct equality is as follows:
>
> \begin{align*}
> (N\mathrm{diag}(w)\mathrm{J}) \odot \mathrm{I} = \mathrm{diag}((N \odot \mathrm{J})w), \quad (N\mathrm{diag}(w)) \odot \mathrm{I} = \mathrm{diag}((N \odot \mathrm{I})w).
> \end{align*}
>
> To clarify:
>
> \begin{align*}
> (N\mathrm{diag}(w)\mathrm{J})\_{i,i} &= \sum_{l,m} N_{i,l}\mathrm{diag}(w)\_{l,m} \mathrm{J}\_{m,i} \newline
> &= \sum_{l} N_{i,l}w_{l} \mathrm{J}\_{l,i} \newline
> &= \sum_{l} N_{i,l} \mathrm{J}\_{i,l}w_{l} \newline
> &= \sum_{l} (N \odot \mathrm{J})\_{i,l}w_l \newline
> &= ((N \odot \mathrm{J})w)\_i,
> \end{align*}
>
> \begin{align*}
> (N\mathrm{diag}(w))\_{i,i} &= \sum_{l} N_{i,l}\mathrm{diag}(w)\_{l,i}  \newline
> &= N_{i,i}w_{i} \newline
> &= \sum_{l} (N \odot \mathrm{I})\_{i,l}w_l \newline
> &= ((N \odot \mathrm{I})w)\_i.
> \end{align*}
>
> We have revised the paper to correct this and to include the above derivations for clarity.

---

> ### Author Response · Authors · 2024-11-21
>
> **Regarding Proof D.1:**
> We appreciate the reviewer’s concern and would like to clarify the context of the proof. When searching for paths in a graph, self-loops do not contribute to a path, since a self-loop would imply a repeated node within the path. Therefore, throughout Appendix D, we assume that the adjacency matrix $A$ has a zero diagonal, which corresponds to the assumption that the graph is simple. Under this condition, equality holds in all the derivations in Appendix D.
>
> The reviewer’s counterexample highlights a valid issue for cases where self-loops are present. In particular, when the diagonal of $A$ is non-zero, it can lead to $\mathrm{J} \odot A = \mathrm{J}$. However, in the case of simple graphs, the diagonal is zero, and hence, $\mathrm{J} \odot A = A$, which preserves the correctness of our derivations.
>
> We have clarified this assumption and its implications in the paper to avoid any further confusion. Thank you again for pointing this out.
>
> **Regarding PDA:**
> The reviewer correctly notes that, in the general case, PDAs serve primarily as language acceptors. However, as the reviewer points out, PDAs can also be used as language generators by reinterpreting their inputs as outputs. To clarify this, we have added an explanation in Section 2 of the paper (“Usually, PDA are language acceptors, but by relabelling the input as output of the PDA, the same PDA can be used as language generators”). Furthermore, the input and output of GramFormer can be understood as the concatenation of the PDA's stack and its output, as described in Section 4 of the paper.
> While we acknowledge that the PDA could be omitted in the design of GramFormer (since CFG and PDA are equivalent), we chose to include it for conceptual clarity. In particular, we believe that the input and output of GramFormer are more naturally aligned with the stack and output of a PDA acting as a generator than with the derivation tree of a CFG.
>
> We have replaced the terms “transition relation” with “transition functions” throughout the paper. We sincerely thank the reviewer for bringing this to our attention.
>
> We hope that our rebuttal effectively addresses the majority of the reviewer’s concerns and provides sufficient clarification to warrant a reconsideration of their evaluation of our paper. We remain available to provide further clarification should it be required.
>
>
>
>
>
>
> [1] Zhao, A., Xu, J., Konaković-Luković, M., Hughes, J., Spielberg, A., Rus, D., & Matusik, W. (2020). Robogrammar: graph grammar for terrain-optimized robot design. ACM Transactions on Graphics (TOG), 39(6), 1-16.
>
> [2] Klinghoffer, T., Tiwary, K., Behari, N., Agrawalla, B., & Raskar, R. (2023). DISeR: Designing Imaging Systems with Reinforcement Learning. In Proceedings of the IEEE/CVF International Conference on Computer Vision (pp. 23632-23642).
>
> [3] Vázquez, H. C., Sanchez, J., & Carrascosa, R. (2023, June). Integrating Hyperparameter Search into Model-Free AutoML with Context-Free Grammars. In International Conference on Learning and Intelligent Optimization (pp. 523-536). Cham: Springer International Publishing.
>
> [4] Vazquez, H. C., Sánchez, J., & Carrascosa, R. (2022, October). GramML: Exploring context-free grammars with model-free reinforcement learning. In Sixth Workshop on Meta-Learning at the Conference on Neural Information Processing Systems.
>
> [5] Bouritsas, G., Frasca, F., Zafeiriou, S., & Bronstein, M. M. (2022). Improving graph neural network expressivity via subgraph isomorphism counting. IEEE Transactions on Pattern Analysis and Machine Intelligence, 45(1), 657-668.
>
> [6] Yun, S., Jeong, M., Kim, R., Kang, J., & Kim, H. J. (2019). Graph transformer networks. Advances in neural information processing systems, 32.

---

> > ### Comment · Reviewer_ZBFH · 2024-11-25
> >
> > Thank you a lot for your response and revision. I have increased my rating.
> >
> > Still, I have a question regarding Thm. A.1: Inductively eliminating $V_c$ and $diag(V_c)$, you show that $G_3$ is at least as expressive as $3$-WL. How do you eliminate $J$ for the backward claim? Or do you even need equivalent expressiveness?

---

> > > ### Author Response · Authors · 2024-11-26
> > >
> > > We sincerely thank you for your thoughtful review and are delighted that you consider our work to be a good paper.
> > >
> > > To address your final question: an explicit equivalence is not required in this case, as we only need $G_3$ to be at least as expressive as 3-WL. However, we can certainly provide the backward proof for completeness.
> > >
> > > We have $M \odot \mathrm{J} = M - M \odot \mathrm{I}$ and $M \mathrm{J}=(M1,\cdots ,M1)−M$. By leveraging Proposition A.1 in the same manner as in the original proof, the operations involving $\mathrm{J}$ are unnecessary. This is because $M1$, $M$, and $M \odot \mathrm{I}$ can all be derived from the initial grammar.

---

### Official Review · Reviewer_XApC · 2024-11-04

**Soundness:** 3
**Presentation:** 3
**Contribution:** 2
**Rating:** 5
**Confidence:** 4

**Summary:**

This paper introduces a grammar-based reinforcement learning (RL) approach to synthesize novel formulae that describe path and cycle counts in graphs. Building on prior work, particularly by Voropaev and Perepechko, which provided matrix multiplication and Hadamard product-based formulae for counting expressions, this study defines a context-free grammar (CFG) capable of generating these expressions. Using a reinforcement learning and transformer-based approach, the authors search for more compact formulae with improved time complexity. They report multiple successes with this framework, though they note that training the model is time-intensive.

**Strengths:**

- The paper addresses a problem of foundational significance, exploring the automation of mathematical creativity through machine learning.
- The approach is well-motivated and effectively presented, with a clear architecture and a detailed explanation of generating formulae from a given grammar.
- The framework successfully recovers several known formulae while also discovering new, more compact ones.

**Weaknesses:**

- The importance of path and cycle counting is not sufficiently argued, particularly for an ICLR audience that may require more context on its broader relevance.
- It remains unclear whether the RL and transformer framework is essential to the solution, as it seems more like a general-purpose tool for exploring terms generated by a given CFG..
- The framework does not provide a proof of correctness for the generated formulae, requiring a time-consuming, manual derivation for each result. Automating the generation of explanations (a sequence of rewriting rules) to demonstrate correctness would significantly improve the approach's appeal.
- The method appears more general beyond its current application domain. The paper could be strengthened by exploring additional applications of the proposed framework.

**Questions:**

1. How are the proofs of equivalence for the grammar-generated formulae derived? Is it challenging to verify correctness once a candidate formula is generated?
2. By characterizing various rewriting rules used in these proofs, could it be possible to automate the generation of more compact formulae using traditional methods, such as a rewriting system?
3. If so, has the proposed approach been compared against exhaustive explorations that utilize such rewriting rules?

---

> ### Author Response · Authors · 2024-11-22
>
> # Response to Reviewer Comments
>
> We would like to thank the reviewer for their thoughtful review. We are pleased that they recognize the foundational significance of the problem addressed in our paper and appreciate our proposed approach. We hope that the following answers will respond to their concerns.
>
> ---
>
> ## **1. First Weakness: importance of path and cycle counting for an ICLR audience**
>
> To address this concern, we have revised the opening paragraph of our introduction to emphasize the potential impact of our findings on the machine learning field, with a particular focus on the geometric learning community. Specifically, we reference the foundational work introducing GSN [1], a GNN framework that incorporates path and cycle counts at the node or edge level into feature representations. This methodology has been shown to significantly enhance the expressive power of GNNs (see Table 1 in [1]). Moreover, this same work highlights the computational cost of such precomputation, underscoring the importance of developing more efficient algorithms for path and cycle counting.
>
> To reinforce the experimental evaluation, we have compared the computation time required by the preprocessing steps of GSN with the use of our formulae. The results, detailed in Appendix D, demonstrate that our formulae significantly outperform the baseline in terms of computational efficiency.
>
> ---
>
> ## **2. Second and fourth Weaknesses: The RL and transformer framework's necessity is unclear, and the method appears more general beyond its current application domain.**
>
> To address these points, this paper demonstrates a specific use case of GRL to highlight its ability to discover novel insights. While the paper primarily focuses on counting paths in undirected simple graphs, inspired by Reviewer 5jnX's Question C4, we extended GRL's application to directed simple graphs. Thus far, GRL has successfully derived formulae for $P_2$, $P_3$, $P_4$, and $P_5$. To the best of our knowledge, these formulae are novel. Notably, these directed graph formulae align with their undirected counterparts, supporting a conjecture of optimality for GRL-discovered formulae, at least in the undirected case.
>
> Additionally, in the context of this new application, we conducted experiments comparing GRL with alternative approaches, such as using MCTS without the assistance of GramFormer and a purely random exploration of the grammar tree. These results, detailed in Appendix E, underline the impact and necessity of our GramFormer component.
>
> Beyond this new use case, we are convinced that the GRL framework is highly generalizable and can be extended to tackle other graph-based computations, such as:
> - **Detecting specific motifs** (e.g., cliques, stars) or patterns with additional constraints (e.g., edge weights, vertex labels).
> - **Matrix-based problems**, including optimization of algebraic operations like matrix multiplication, tensor factorization, or Boolean satisfiability problems.
> - **Domain-specific applications**, such as network analysis (e.g., communication networks, social networks) or biology (e.g., protein interaction networks, metabolic pathways).
>
> ---
>
> ## **3. Third Weakness: The framework lacks a proof of correctness for the generated formulae, requiring manual derivation for each result.**
>
> We address this point through the reviewer’s specific questions
> ---
>
> ## **4. First Question :**
>
>    The equivalence proofs were conducted manually, while the correctness of the formulae is directly ensured by GRL. Specifically, if all possible walks of length $l$ are represented within the graphs of the dataset used for GRL, the outputs of GRL inherently validate the correctness of the formulae. When GRL produces a set of sentences with a maximal value, it guarantees that this set accurately counts the $l$-paths or cycles. Furthermore, the successful derivation of equivalence proofs demonstrates that GRL's results are fully interpretable and verifiable by humans.
>
> ---
>
> ## **5. Second and Third Question :**
>
> We acknowledge the reviewer's observation that a rewriting system could potentially be effective; however, due to our limited expertise in this area, we cannot confidently assert its applicability in this specific context. Additionally, for the directed case or for paths of length greater than 7, no existing formulae are available for reduction, rendering a rewriting system inapplicable.
>
> ---
>
> ## **Conclusion**
>
> We hope that our rebuttal effectively addresses the majority of the reviewer’s concerns and provides sufficient clarification to warrant a reconsideration of their evaluation of our paper. Thank you again for your valuable feedback.
>
> ## References
>
> [1] Bouritsas, G., Frasca, F., Zafeiriou, S., & Bronstein, M. M. (2022). Improving graph neural network expressivity via subgraph isomorphism counting. *IEEE Transactions on Pattern Analysis and Machine Intelligence, 45*(1), 657-668.

---

> > ### Comment · Reviewer_XApC · 2024-12-02
> >
> > Thank you for your response and revision.

---

### Official Review · Reviewer_LTAB · 2024-11-05

**Soundness:** 3
**Presentation:** 3
**Contribution:** 2
**Rating:** 6
**Confidence:** 4

**Summary:**

This paper proposes a Reinforcement Learning based method to discover an algebraic formula involving (adjacency) matrix of an undirected graph and some constant matrices, for counting the number of paths and cycles of specific (short) lengths in the graph. The main experimental achievement is finding (simple) formulas for path lengths l = 2, 3, 4, 5, 6. In the case of l = 2 the discovered formula is equal to the best known formula proposed by Voropaev and Perepechko (2012), and for l = 3, 4, 5, 6 the algorithm found even more efficient alternative expressions than those introduced by Voropaev and Perepechko.

The algorithm implements a searching strategy to find an optimal formula generated by an appropriate context-free grammar (CFG), or equivalently by a Pushdown Automaton (PDA). To this aim the authors propose a Monte Carlo Tree Search (MCTS) based Deep Reinforcement Learning algorithm, termed Grammar Reinforcement Learning (GRL).

**Strengths:**

Counting the number of paths and cycles (of specific lengths) in graphs is an important task in algorithmics and combinatorics with many applications to different fields. The authors show that using the DRL approach in combination with the Monte Carlo Tree Search method allows the discovery of more efficient matrix-based formulae for counting paths (of lengths up to six) than the best known so far. This is a nice achievement of the work.

The authors present MCTS-based DRL algorithm, termed Grammar Reinforcement Learning (GRL), that started with context-free grammar uses an equivalent Pushdown Automaton (PDA) for generating searching trees. To learn policy and value functions in the GRL a transformer architecture is proposed which models the PDA. This provides an interesting connection between the use of grammars, PDAs and reinforcement learning. Using this approach, the algorithm were able to discover formulae that are more efficient than those proposed by Voropaev and Perepechko (2012).

**Weaknesses:**

The paper is largely based on the work of Piquenot et al. (ICLR 2024), that introduced a methodology based on CFGs that led to the construction of a new Grammatical Graph Neural Network model. It is provably 3-WL equivalent. As stated in the submitted work the generative framework of Piquenot et al. (ICLR 2024) already produces all formulae identified by Voropaev and Perepechko (2012). Hence, the innovative aspects of this submission are somewhat limited. Furthermore, it is not clear to what extent the proposed approach is generic. It would be interesting to have a discussions on such applications. It would be also interesting to compare the methods using the obtained matrix-based formulae with other methods for counting paths and cycles of lengths up to six.

**Questions:**

Please discuss the issues mentioned above.

L. 134: in the formula brackets \{...\} are missing.

L. 186: How do you define the adjacency matrix A? Do you mean really that A_{i,j} = 1 iff i is connected with j in G?

L. 814: Lemme D.1 --> Lemma D.1

---

> ### Author Response · Authors · 2024-11-22
>
> # Response to Reviewer Comments
>
> We thank the reviewers for their thorough, positive and constructive feedback. We are pleased to see that the reviewers believe that our paper addresses an important problem in algorithmics and combinatorics with many applications to different fields, and that our proposal allows nice achievements.
>
> Below, we address the specific points raised in the review.
>
> ---
>
> ## **1. First Weakness: Positioning w.r.t. the Work of [1]**
>
> The reviewer is absolutely correct that our submission builds upon the work of [1], as well as [2]. Both of these works laid the foundation for grammatical approaches to designing Graph Neural Networks. However, it is not accurate to state that the work of [1] already produces all formulae identified by [3]. To the best of our understanding, this was not the purpose of their work.
>
> While [1] provide a strong foundation for grammatical approaches, our work extends their methodology by focusing specifically on **algorithm design** through reinforcement learning. This focus enables us to explore novel formulae beyond those identified in prior work. Notably, the formulae discovered for path lengths $P_3, P_4, P_5, P_6$ are more efficient than the best known results to date.
>
> We believe this misunderstanding stems from a confusing sentence in our initial submission. Specifically, we wrote:
> > “In order to build a 3-WL GNN, Piquenot et al. (2024) introduced a 3-WL Context-Free Grammar (CFG), a generative framework that notably produces all formulae identified by Voropaev and Perepechko (2012).”
>
> Our intention with this sentence was to highlight that the generative framework proposed by Piquenot et al. could, with minor modifications, produce the formulae introduced by [3]. To clarify this, we have revised the sentence in the manuscript as follows:
> > “We observed that, with minor modifications, this generative framework is capable of producing all the formulae previously identified by Voropaev and Perepechko (2012).”
>
> We sincerely apologize for the ambiguity in the original wording and hope this clarification resolves the misunderstanding.
>
> ---
>
> ## **2. Second Weakness: Genericity of the Proposed Approach**
>
> We appreciate the reviewer’s observation regarding the need to further discuss the genericity of our framework. To better illustrate its potential, we have added a new use case during the rebuttal period. This use case involves counting paths in **directed simple graphs**, requiring modifications to the grammar to account for the transposition operation (see Appendix E for details). Using this adapted grammar, GRL successfully derived formulae for $P_2, P_3, P_4,$ and $P_5$, which, to the best of our knowledge, do not exist in the literature. Interestingly, these formulae match their counterparts for undirected graphs, providing evidence supporting the conjecture that the GRL-discovered formulae may be optimal in both settings.
>
> Additionally, in the context of this new application, we conducted experiments comparing GRL with alternative approaches, such as using MCTS without the assistance of GramFormer and a purely random exploration of the grammar tree. These results, detailed in Appendix E, underline the impact and necessity of our GramFormer component.
>
> Beyond this new use case, we are convinced that the GRL framework is highly generalizable and can be extended to tackle other graph-based computations, such as:
> - **Detecting specific motifs** (e.g., cliques, stars) or patterns with additional constraints (e.g., edge weights, vertex labels).
> - **Matrix-based problems**, including optimization of algebraic operations like matrix multiplication, tensor factorization, or Boolean satisfiability problems.
> - **Domain-specific applications**, such as network analysis (e.g., communication networks, social networks) or biology (e.g., protein interaction networks, metabolic pathways).
>
> While our current focus is on foundational path and cycle counting, we recognize the importance of empirically validating the generality of GRL in broader applications. We plan to explore these extensions in future work.
>
> ---
>
> ## **3. Third Weakness: Comparison with Other Methods for Path Counting**
>
> To address this weakness, as well as the third weakness raised by reviewer ZBFH regarding the significance of path and cycle counting, we have compared the computation time required by a classical algorithm for computing paths and cycles with the use of our formulae. Specifically, we take as a baseline the preprocessing steps of GSN, a GNN framework proposed by [4], which incorporates path and cycle counts at the node or edge level into feature representations.
>
> The results, detailed in Appendix D, demonstrate that our formulae significantly outperform the baseline in terms of computational efficiency. This comparison highlights the practical benefits of our contributions, particularly in scenarios where efficiency is critical.

---

> > ### Author Response · Authors · 2024-11-22
> >
> > ## **4. First and Third Question: Typos**
> >
> > Both typos have been corrected in the revised version.
> >
> > ---
> >
> > ## **5. Second Question: Adjacency Matrix Definition**
> >
> > The reviewer is fully right. It is the classical definition of an adjacency matrix.
> >
> > ---
> >
> > ## **Conclusion**
> >
> > We sincerely thank the reviewers for their thoughtful comments, which have allowed us to improve the clarity, scope, and presentation of our work. In this revised version, we have addressed the concerns raised by:
> > - Clarifying our positioning with respect to prior work by [1].
> > - Demonstrating the genericity of our framework by including a new use case on directed simple graphs, which highlights the potential of GRL to generate novel formulae.
> > - Comparing our method to state-of-the-art approaches to underline the computational benefits of the formulae discovered by GRL.
> > - Correcting ambiguities and addressing the questions and concerns raised.
> >
> > We hope the revised manuscript addresses their concerns fully.
> >
> >
> > [1] Piquenot, J., Moscatelli, A., Berar, M., Héroux, P., Raveaux, R., RAMEL, J. Y., & Adam, S. (2024). G $^ 2$ N $^ 2$: Weisfeiler and Lehman go grammatical. In The Twelfth International Conference on Learning Representations.
> >
> > [2] Geerts, F. (2021). On the expressive power of linear algebra on graphs. Theory of Computing Systems, 65, 179-239.
> >
> > [3] Voropaev, A. N., & (2009)Perepechko, S. N. The number of fixed length cycles in an undirected graph. Explicit formulae in case of small lengths. Mathematical Modeling and Computational Physics (MMCP2009), 148.
> >
> > [4] Bouritsas, G., Frasca, F., Zafeiriou, S., & Bronstein, M. M. (2022). Improving graph neural network expressivity via subgraph isomorphism counting. IEEE Transactions on Pattern Analysis and Machine Intelligence, 45(1), 657-668.

---

> > > ### Comment · Reviewer_LTAB · 2024-11-26
> > > **Comment**
> > >
> > > Thank you for clarifying my concerns and improving presentation. I have no further questions.

---

> > > > ### Author Response · Authors · 2024-11-29
> > > >
> > > > We are delighted that our responses have satisfied the reviewer, and are convinced that they have improved the quality of the paper. We remain available to clarify other points if necessary.

---

### Meta-Review · Area_Chair_QC4T · 2024-12-20

**Metareview:**

The paper presents Grammar Reinforcement Learning (GRL) that combines MCTS with a transformer architecture that models a Pushdown Automaton within a context-free grammar (CFG) framework. GRL is empirically shown to discover new matrix-based formulas for path/cycle counting that improve
computational efficiency by factors of two to six. One reviewer pointed out that it appears that previous work could have already achieved the same results; the authors, however, pointed out that this is not the case and changed the wording. Another reviewer points out that GRL does not prove the generated formulae's correctness, requiring a time-consuming, manual derivation for each result. While I agree that automating the generation of explanations (a sequence of rewriting rules) to demonstrate correctness would significantly improve the approach's appeal, the first step is to show that it is useful. Moreover, there was an active discussion between one reviewer and the authors, fixing an error in the presented proof. I very much appreciate this, and the reviewer, in the end, raised the rating. So, while I fully agree that the experimental evaluation could be broadened, the discovery is interesting and will motivate others to push for AI-based algorithm discovery. Hence, I suggest going with the overall assessment of the reviews and accepting the paper.

**Additional Comments On Reviewer Discussion:**

The discussion arose from issues raised in the reviews. There were not many iterations, except for a very interesting discussion that led to fixing a bug in the proofs. This has helped in turn to accept the paper.

---

### Decision · Program_Chairs · 2025-01-22

Accept (Poster)